# Ligand-induced shifts in conformational ensembles that describe transcriptional activation

**Sabab Hasan Khan[1], Sean M Braet[2], Stephen John Koehler[2], Elizabeth Elacqua[2], Ganesh Srinivasan Anand[2], C Denise Okafor[1,2]\***

[1]Department of Biochemistry and Molecular Biology, Pennsylvania State University, State College, United States; [2]Department of Chemistry, Pennsylvania State University, State Park, United States

**Abstract** Nuclear receptors function as ligand-regulated transcription factors whose ability to regulate diverse physiological processes is closely linked with conformational changes induced upon ligand binding. Understanding how conformational populations of nuclear receptors are shifted by various ligands could illuminate strategies for the design of synthetic modulators to regulate specific transcriptional programs. Here, we investigate ligand-induced conformational changes using a reconstructed, ancestral nuclear receptor. By making substitutions at a key position, we engineer receptor variants with altered ligand specificities. We combine cellular and biophysical experiments to characterize transcriptional activity, as well as elucidate mechanisms underlying altered transcription in receptor variants. We then use atomistic molecular dynamics (MD) simulations with enhanced sampling to generate ensembles of wildtype and engineered receptors in combination with multiple ligands, followed by conformational analysis and correlation of MD-based predictions with functional ligand profiles. We determine that conformational ensembles accurately describe ligand responses based on observed population shifts. These studies provide a platform which will allow structural characterization of physiologically-relevant conformational ensembles, as well as provide the ability to design and predict transcriptional responses in novel ligands.

**\*For correspondence:**
cdo5093@psu.edu

**Competing interest:** The authors declare that no competing interests exist.

## Editor's evaluation

This paper reports a fundamental set of results describing the activation of nuclear receptors. The evidence to support the relationship between function and ligand-induced shift in the conformational ensembles is based on a compelling combination of experimental and computational approaches. The manuscript has implications for fully understanding how perturbation of the conformational ensembles of proteins, in general, orchestrates function. The findings will be of interest to a broad audience in biochemistry and structural, molecular, and evolutionary biology.

## Introduction

Nuclear receptors (NRs) are master regulators of diverse physiological functions, including reproduction, inflammation, development, and metabolism (*Francis et al., 2003*; *Lee et al., 2008*; *Rhen and Cidlowski, 2005*; *Puengel et al., 2022*; *Sun et al., 2021*; *Connelly et al., 2022*; *Tenbaum and Baniahmad, 1997*). The ability of this class of ligand-activated transcription factors to control critical cellular functions is driven by binding of lipophilic ligands. Members of the NR superfamily include steroid hormone receptors, such as the well-known estrogen and progesterone receptors, which are regulated by cholesterol-derived hormones. NRs share a characteristic modular domain architecture,

including a highly conserved DNA binding domain and a moderately conserved ligand binding domain (LBD) which houses a hydrophobic binding cavity. In steroid receptors, ligand binding induces a conformational change in the receptor, followed by dissociation from chaperone proteins (*Fang et al., 2006*), binding to the DNA response elements located in promoter regions of target genes (*Freedman, 1992*; *Bagchi et al., 1988*; *Mangelsdorf et al., 1995*; *Becker et al., 1986*), and recruitment of coregulator proteins (*Tsai et al., 1988*; *Nettles and Greene, 2005*; *McKenna et al., 1999*).

Differential conformational changes induced by ligands permit the recruitment of coregulator proteins that either promote transcriptional activation, or repression of target genes (*Nettles and Greene, 2005*; *Beekman et al., 1993*; *Glass and Rosenfeld, 2000*). Activating ligands (agonists) will switch the receptor into a so-called active state, generally defined by the conformation in which the C-terminal helix (i.e. helix 12) is packed against the LBD, stabilized by interactions with nearby helices 3 and 4, as well as ligand contacts (*Nichols et al., 1998*; *Nettles et al., 2007*; *Weikum et al., 2017*; *Poujol et al., 2000*). This region comprises the activation function 2 (AF-2) surface (*Heery et al., 1997*). The active state positioning of H12 allows coactivator proteins to be recruited to the LBD (*Li et al., 2003*; *Shang et al., 2020*; *Bledsoe et al., 2002*). Conversely, repositioning or destabilization of H12 is associated with inactive states of receptors such as an apo state or an antagonist-bound state (*Nichols et al., 1998*; *Min et al., 2021*). However, experimental evidence indicates that NRs exist as a dynamic ensemble of conformations whose populations can be modulated by ligand binding or other perturbations (*Kojetin and Burris, 2013*). While it is an ongoing challenge to structurally characterize NR conformational ensembles and reveal ligand-induced population shifts, experimental methods such as solution state NMR have enabled great advances, revealing how ligands of diverse efficacy and potency affect the active state (*Chrisman et al., 2018*; *Shang et al., 2019*).

Powerful advancements in computational approaches have increased their application for the study of protein conformational ensembles. Computational methods for conformational sampling are notoriously hampered by two major challenges: inherent limitations in forcefields, and the difficulty of achieving sufficient sampling of the free energy landscape (*Allison, 2020*; *Thomasen and Lindorff-Larsen, 2022*). Enhanced sampling methods applied to molecular dynamics (MD) simulations, including accelerated MD, metadynamics and replica exchange have been useful for overcoming limitations in studying conformational ensembles, providing physical descriptions that illuminate structural and functional protein mechanisms (*O'Hagan et al., 2020*; *Araki et al., 2021*; *Kappel et al., 2015*; *Nagpal et al., 2020*; *Motta et al., 2018*; *Wang et al., 2012*). Previously we showed that conformational ensembles of NRs generated by accelerated MD simulations underwent shifts upon addition of ligands (*Okafor et al., 2020*). Unexpectedly, the population shifts correlated with the transcriptional activity of the ligands. Thus, understanding the effects of ligands on NR ensembles can be a promising approach for screening and predicting functional profiles of new NR ligands.

In this work, we expand our previous system by characterizing conformational shifts in a set of engineered receptors with altered ligand specificity and transactivation potential. We investigate conformational ensembles using a reconstructed ancestral steroid receptor, AncSR2. In transcriptional assays, AncSR2 was activated by 3-ketosteroid hormones, that is steroids with a non-aromatized A-ring and a keto substituent at the carbon 3 position, while remaining unresponsive to hormones with an aromatic A-ring, i.e., estrogens (*Figure 1A*; *Eick et al., 2012*). To produce a diverse set of receptors with a range of functional profiles, we created four AncSR2 variants by mutating Methionine 75 (M75), a critical residue located on helix 5 (H5) of the LBD that is conserved across modern steroid receptors and shown by us and others (*Okafor et al., 2020*; *Harms et al., 2013*) to be crucial for hormone recognition. We use cellular assays to characterize transcriptional activity of the variants with diverse ligands, followed by biophysical and structural analyses to dissect the structure-activity criteria underlying observed functional responses. We then use MD simulations to predict the conformational effects of M75 substitutions, generate conformational ensembles and observe population shifts that occur upon binding to aromatized and non-aromatized hormones. Finally, we correlate experimental results with shifts in computational ensembles to elucidate ligand-induced effects in a set of receptors.

We observed that the M75 mutations achieved a range of functional profiles in AncSR2, including constitutively active and completely inactive states, enabling a broad investigation into how receptor function affects population shifts within NR ensembles. Our studies reveal that ensembles generated are sensitive to both M75 mutations and ligand identity. In AncSR2 receptor variants, population shifts, assessed by clustering ligand-bound conformations with unliganded ensembles, correlated

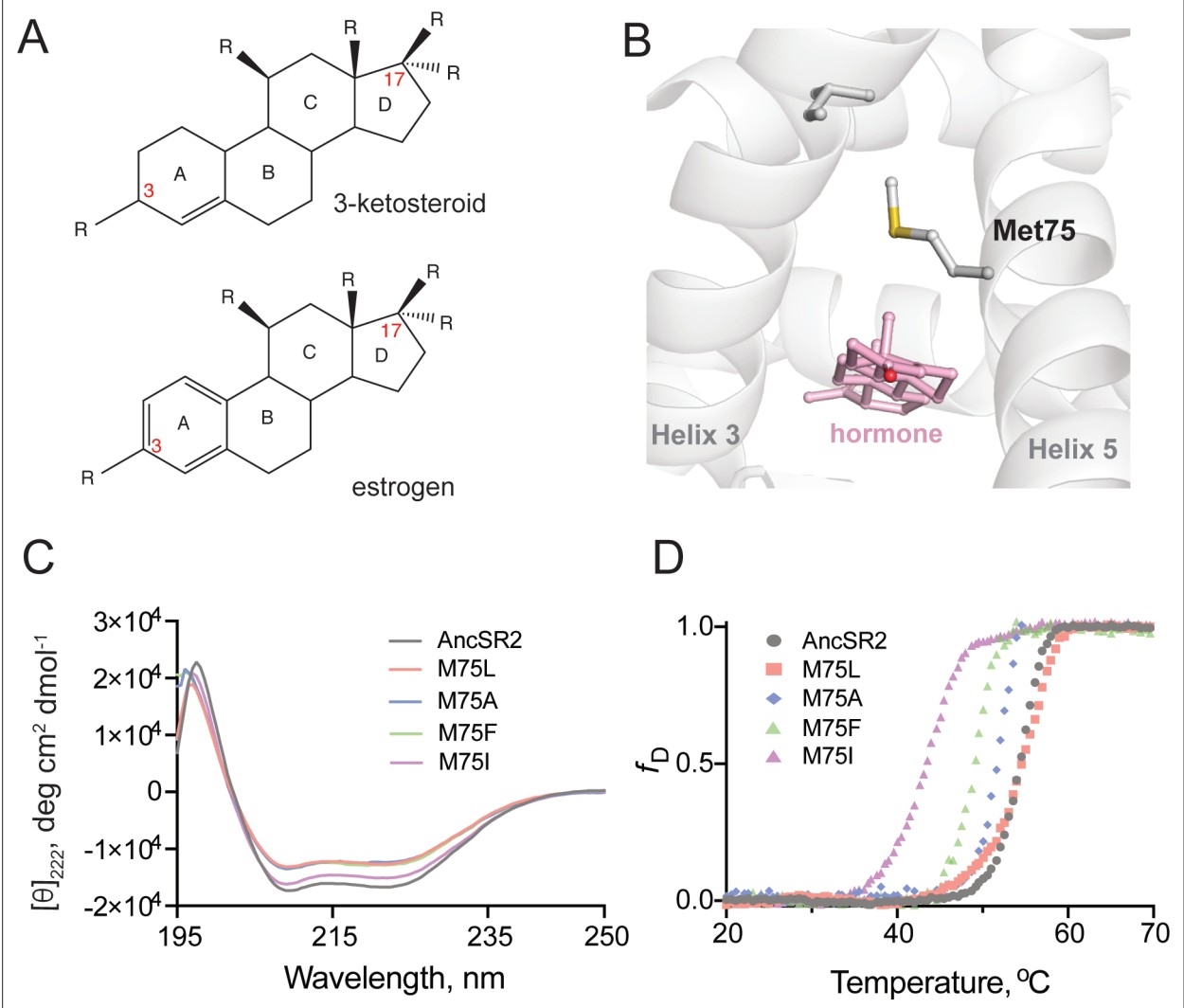

**Figure 1.** Structure and stability of AncSR2 and M75 mutants. (**A**) Chemical structure of A-ring aromatized estrogen and non-aromatized 3-ketosteroid. (**B**) Met75 (H5) is positioned to form critical contacts with H3 and the hormone located in the ligand binding pocket. (**C**) Far-UV CD spectra of AncSR2 and its variants in the wavelength range 195–250 nm, showed mutants LBD remains in the folded state similarly as AncSR2. (**D**) Normalized heat-induced denaturation transition curves of AncSR2 and its variants monitored by change in the $[\theta]_{222}$ as function of temperature. Each curve represents averaged measurements from two replicates and two independent purifications.

The online version of this article includes the following source data and figure supplement(s) for figure 1:

**Figure supplement 1.** Purity and homogeneity of mutants were assessed by SDS-PAGE and Gel filtration chromatography.

**Figure supplement 1—source data 1.** 14% SDS-PAGE showing purity of the purified LBD of AncSR2 and variants.

with functional properties of ligands. Changes in ensemble populations were predictive of strong, weak or no agonist activity in ligands. To reveal the origin of inactivity or constitutive activity in AncSR2 variant, we employed hydrogen deuterium exchange mass spectrometry and ligand binding assays. These studies confirm the inherent promise in this MD-based approach for characterizing diverse NR-ligand ensembles.

## Results

### Substitution of M75 has minor effects on global structure and larger impact on local interactions

Met75, located on helix 5 (H5) of AncSR2 holds structural, functional, and evolutionary significance for steroid receptors. Notably, M75 engages helix 3 (H3) residues via van der Waals contact (*Figure 1B*), an interaction that is conserved in extant glucocorticoid, mineralocorticoid, and progesterone receptors (*Zhang et al., 2005*). M75 was also shown to interact with bound hormones (*Harms et al., 2013*) representing an ideal position for mutagenesis to create a series of engineered receptors with altered potency and ligand specificity. We generated M75L, M75I, M75F, M75A mutants of AncSR2 and performed biophysical characterizations to ensure that mutations do not substantially impact structure and stability. Wildtype (WT) AncSR2 LBD and mutants were expressed and purified to homogeneity (*Figure 1—figure supplement 1A*). Gel filtration profiles show that similar to WT, mutant LBDs elute as a single peak, suggesting that mutations do not affect the globular nature of the protein (*Figure 1—figure supplement 1B-C*). We used circular dichroism spectroscopy (CD) to determine the impact of M75 mutations on the structure of AncSR2. Far-UV CD (195–250 nm) spectral measurements of the WT AncSR2 and mutants reveal features characteristic of α-helical proteins, that is, negative minima at 208 nm and 222 nm and positive maximum around 190 nm (*Figure 1C*). However, a slight decrease in the mean residue ellipticity at both negative minima was observed in mutants (M75L, M75A, M75F) as compared to WT AncSR2. Overall, CD measurements confirm that mutations do not affect the global secondary structure of AncSR2.

We also tested the effect of mutations on the stability of AncSR2 and M75 variants by following changes in the CD signal at 222 nm as a function of temperature (*Figure 1D*). The equilibrium denaturation curves for each protein were analyzed to obtain melting temperatures ($T_m$). The apparent $T_m$ values for M75L mutant and WT are identical, within experimental error. The apparent $T_m$ values of M75A, M75F and M75I are respectively 2.3, 5.2, and 10.9 °C lower than AncSR2. Thus while mutant receptors retain secondary structural characteristics of WT AncSR2, stability is reduced in a few variants which may be reflective of local, structural effects.

### Transcriptional responses in M75 variants span a broad activity spectrum

To characterize transcriptional activity in engineered AncSR2 variants, we measured transactivation in cell-based luciferase reporter assays using five hormones: four 3-ketosteroids (progesterone, DHT, hydrocortisone, aldosterone) and estradiol (Structures shown in *Figure 2—figure supplement 1*). All 3-ketosteroid hormones activate AncSR2 with $EC_{50}$ in the sub-nanomolar range except DHT which had a nanomolar $EC_{50}$, consistent with earlier reports (*Eick et al., 2012*; *Figure 2A*). As previously observed, estradiol is not able to activate the receptor (*Figure 2A and G*). In the M75 mutants, we observe a wide range of functional behavior. The M75A variant largely recapitulates the activity profile of WT with the largest differences being loss of DHT activation and reduced efficacy ($E_{max}$) in aldosterone (*Figure 2B*). Potency is reduced for all ligands in M75F activation (*Figure 2C*), with no activity observed in estradiol and DHT. Progesterone activation in M75F is reduced by ~3 orders of magnitude relative to WT AncSR2.

None of the hormones activated M75I (*Figure 2E*), suggesting that this mutation may inhibit ligand binding. Strikingly, while the M75L variant is the only receptor activated by aromatized and non-aromatized hormones, the efficacies of the hormones are drastically reduced ($E_{max}$ ~2–5) (*Figure 2D*) compared to AncSR2 ($E_{max}$ = 11–16) (*Figure 2B*). We then assayed transactivation in the absence of ligands for all AncSR2 variants and confirmed the presence of basal activity in the M75L mutant (*Figure 2F*) that is independent of the cell line used (*Figure 2—figure supplement 2A*). To assess whether transcriptional responses observed in M75I and M75L variants are caused by differential expression of these receptors, we performed western blots and observed that mutant proteins are expressed at levels comparable to WT AncSR2 (*Figure 2—figure supplement 2B*).

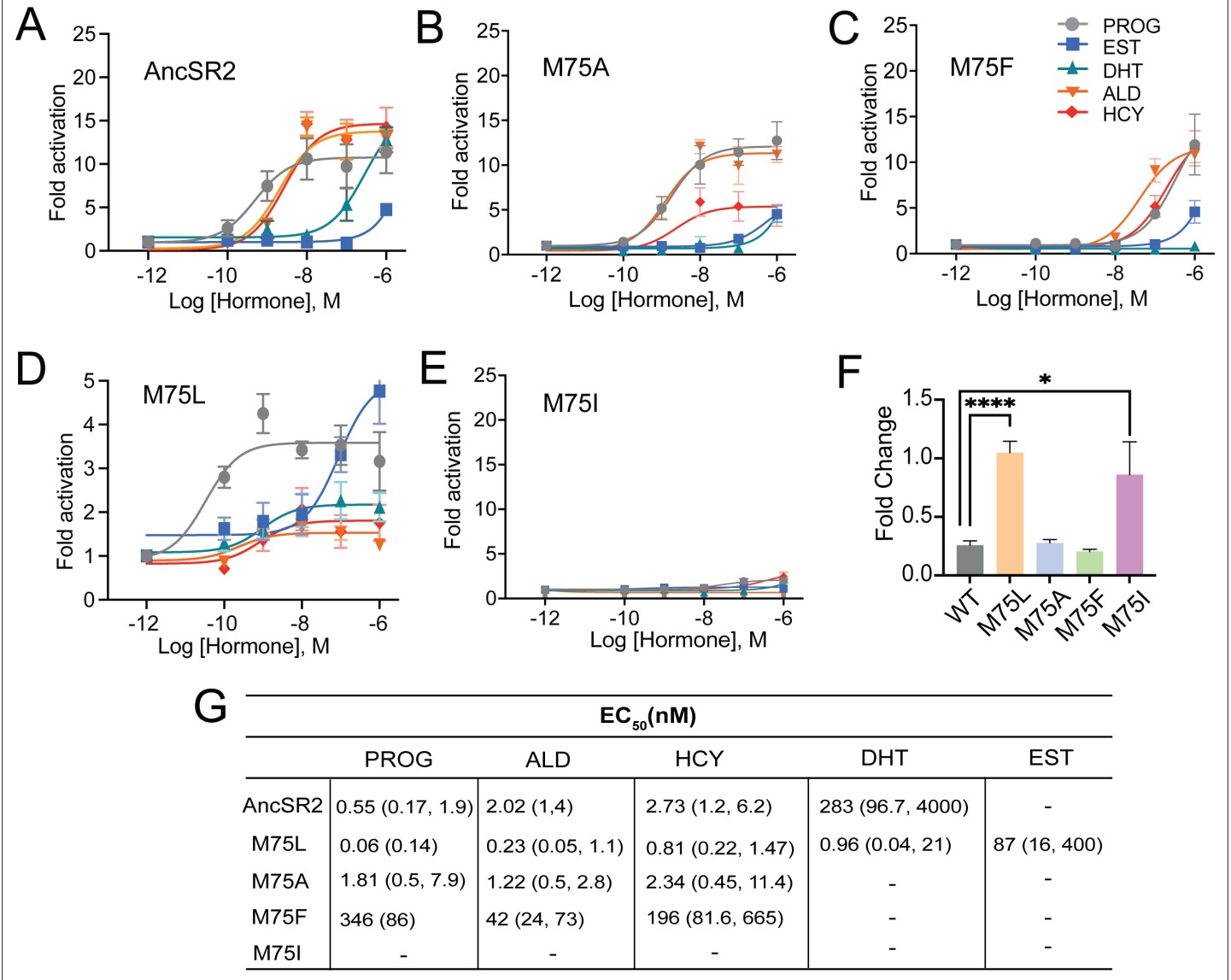

**Figure 2.** Differential ligand activation of AncSR2 variants. Dose-response curves of AncSR2 and its variants in the presence of aromatized and non-aromatized hormones. Five hormones are used: progesterone (PROG), aldosterone (ALD), hydrocortisone (HCY), Dihydrotestosterone (DHT) and Estradiol (EST). (**A**) AncSR2 receptor showed differential response to non-aromatized hormones with no response to estradiol with ligand treatment up to 1 μM. (**B**) M75A substitution slightly increased the fold activation as compared to AncSR2 with no change in the receptor efficacy for progesterone, aldosterone, hydrocortisone. (**C**) M75F substitution decreased the receptor responsiveness for hormones. (**D**) M75L receptor efficacy is significantly reduced compared to WT AncSR2. In contrast to WT AncSR2, M75L was activated by estradiol. (**E**) M75I substitution completely abolished the ligand activation for both types of hormones. Each data point is an average of two to three biological replicates. The error bar associated with each data point represent SEM. (**F**) M75L and M75I receptors fold change over empty vector in the absence of hormone suggest that they exhibit constitutive activity. Two-tailed unpaired t-test, (****) $p<0.0001$, (*) $p<0.05$. (**G**). Table shows $EC_{50}$ values obtained from the analysis of dose-response curves of individual receptors for different hormones. 95% confidence interval values are shown in parentheses.

The online version of this article includes the following source data and figure supplement(s) for figure 2:

**Figure supplement 1.** Chemical structures of different hormones used in the FP-based competition binding experiments.

**Figure supplement 2.** Constitutive activity and expression levels of AncSR2 variants.

**Figure supplement 2—source data 1.** Western blot experiments to show that expression of mutant proteins are comparable to WT AncSR2.

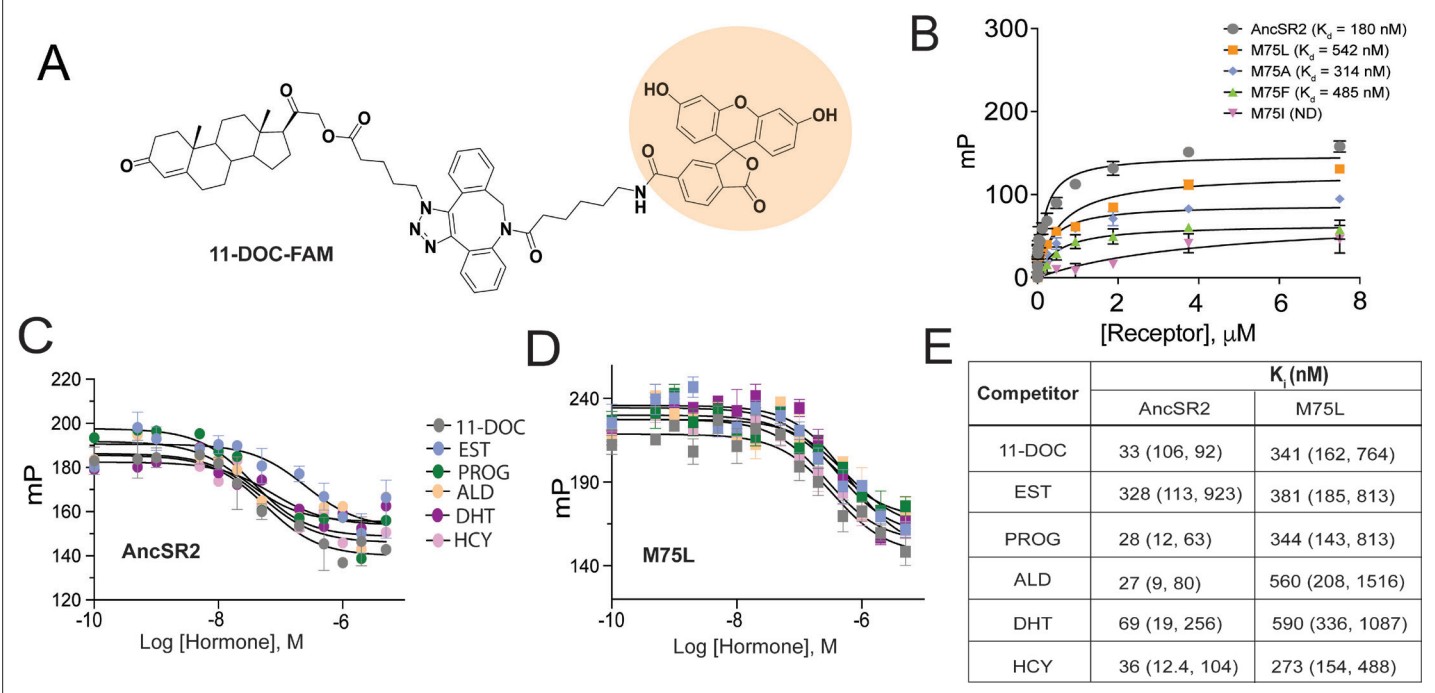

**Figure 3.** Fluorescence polarization assay of binding affinity of AncSR2 and M75 mutants. (**A**) Structure of synthesized probe: FAM labelled 11-DOC (11-DOC-FAM). (**B**) 11-DOC-FAM binds to AncSR2-LBD and M75L, M75A, M75F with equilibrium dissociation constant, $K_d$ = 180 nM (107, 293) and $K_d$ = 542 nM (330,869), $K_d$ = 314 nM (127, 692), $K_d$ = 485 nM (227, 986), respectively. 95% confidence intervals are shown in parentheses. Binding data is the average of six (for WT AncSR2 and M75L) and three (M75A, M75F and M75I) replicates from two and one independent experiment, respectively where each experiment consists of three independent replicates. (**C** and **D**) FP-based competition binding experiments shows that all steroid hormones bind WT AncSR2 and M75L with nM inhibition constants ($K_i$), respectively. Error bars indicate SD from three independent replicates. (**E**) $K_i$ values obtained for five hormones from FP-based competition ligand binding assay. 95% confidence intervals from independent triplicate measurements are shown in parentheses.

## Ligand binding and overall AncSR2 dynamics are impacted by M75 mutations

To investigate the molecular basis for the transactivation profiles of AncSR2 variants, particularly the unexpected results observed in M75I and M75L mutants, we developed a binding assay to probe hormone binding to AncSR2 and all variants. For this purpose, we have designed a probe by linking 11-deoxycorticosterone (11-DOC), a potent AncSR2 agonist (*Eick et al., 2012*) to fluorescein (FAM). By titrating purified AncSR2 LBD against a fixed concentration of 11-DOC-FAM (*Figure 3A*), we obtained a saturation binding curve with an equilibrium dissociation constant $K_d$ = 180 nM (*Figure 3B*). To validate that 11-DOC-FAM binds the AncSR2 ligand binding pocket, we used a competition assay to measure the $K_i$ (inhibition constant) of unlabeled 11-DOC. We observe that unlabeled 11-DOC outcompeted the 11-DOC-FAM with $K_i$ = 33 nM, approximately five-fold lower than $K_d$ (*Figure 3C*). Similarly, previous fluorescent probes for SRs have been reported with 10-fold lower $K_d$ compared to the unlabeled ligand (*Blommel et al., 2004*). Using the same assay, we determined the $K_d$s for M75L, M75A, M75F, and M75I variants (*Figure 3B*). No binding is observed in M75I while all other variants display reduced binding affinity compared to AncSR2.

We then used the competition assay to determine $K_i$s of the five aforementioned hormones for AncSR2 (*Figure 3C*). With varying affinities, all ligands are able to outcompete 11-DOC-FAM from the AncSR2 binding pocket (*Figure 3E*). The $K_i$ values for progesterone, hydrocortisone and aldosterone only differed slightly, but were lower than those observed for DHT and the aromatized hormone, estradiol (*Figure 3E*). Thus, AncSR2 binds 3-ketosteroids preferentially over aromatized hormones, with our results suggesting that a C17 acetyl substituent in hormones may confer a binding advantage. To identify a rationale for the constitutive activity in M75L, we used the same assay to quantify binding of steroid hormones. We observe that in addition to reduced affinity for 11-DOC-FAM compared to WT AncSR2 (*Figure 3B*), the M75L substitution also reduces affinity for all steroid hormones (*Figure 3D*

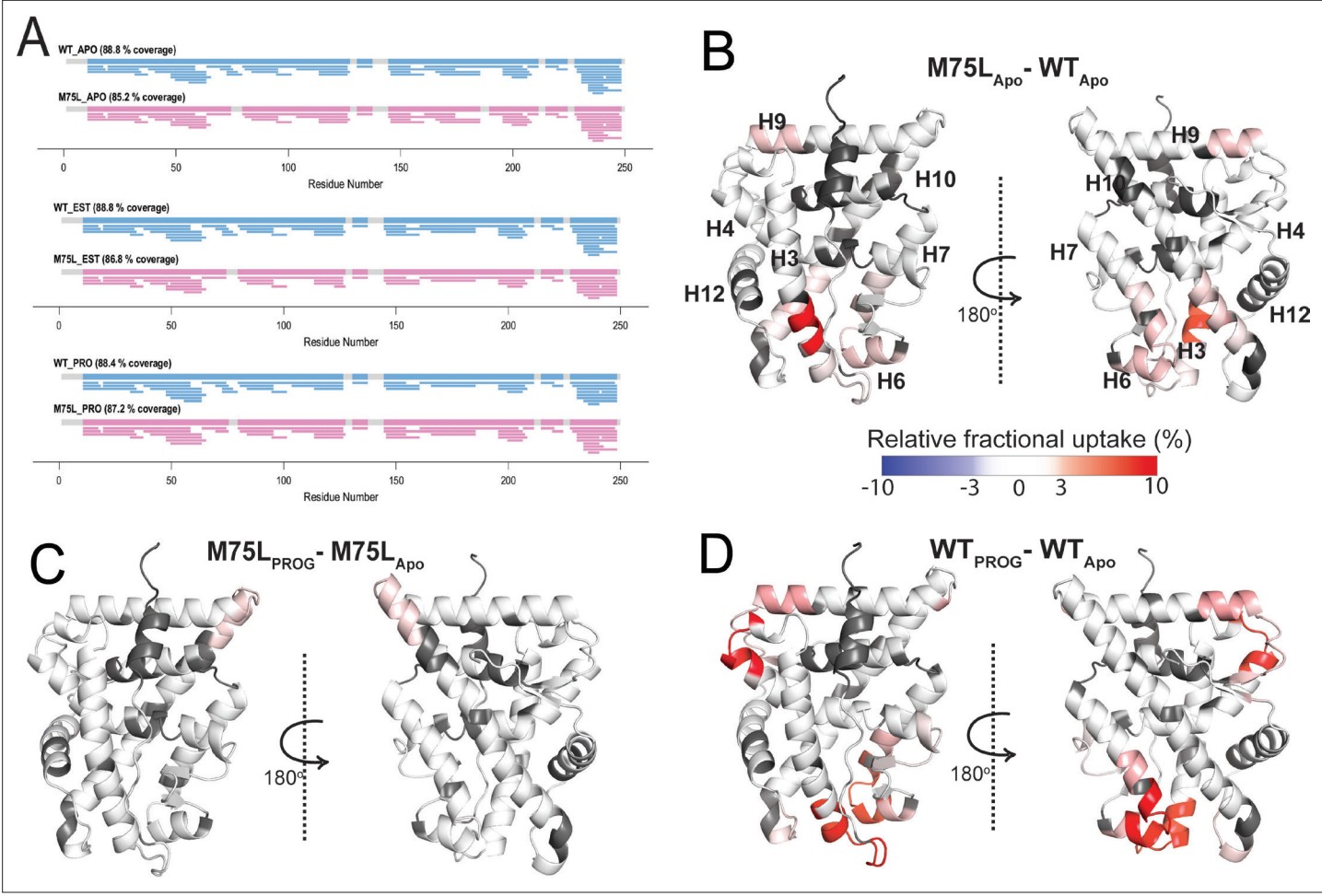

**Figure 4.** Conformational effects of AncSR2 M75L mutation probed by HDX-MS. (**A**) Sequence coverage maps for WT AncSR2 (blue) and M75L (pink) in their apo, estradiol (EST), and progesterone (PROG) bound form. (**B**) The percentage difference in relative fractional uptake (ΔRFU) of deuterium at a time point (15 min) between M75L and WT AncSR2 apo states (M75L$_{Apo}$ – WT$_{Apo}$). (**C**) ΔRFU between M75L-progesterone bound and M75L Apo state (M75L$_{PROG}$-M75L$_{Apo}$). (**D**) ΔRFU between WT-progesterone bound and WT Apo state (WT$_{PROG}$-WT$_{Apo}$). Color bar indicates the fractional difference in relative deuterium uptake for the two states compared. Positive percentage numbers (red) correspond to higher deuterium exchange in state A compared to state B, that is, deprotection, while negative numbers (blue) indicate lower exchange, that is, protection against deuteration exchange. Dark grey regions represent peptides with no sequence coverage.

The online version of this article includes the following figure supplement(s) for figure 4:

**Figure supplement 1.** Deuterium exchange difference plot (Woods plot) between M75L and WT AncSR2 apo states (M75L$_{Apo}$ – WT$_{Apo}$) at multiple time points.

**Figure supplement 2.** Deuterium exchange difference plots (Time = 15 min) between M75L and AncSR2 apo and ligand-bound states.

**Figure supplement 3.** The relative fractional uptake of AncSR2 and M75L under different states.

**Figure supplement 4.** Relative fractional uptake of M75L apo state with M75L-EST (**A**) and M75L-PRO (**B**) bound state.

*and E*). None of the ligands bind M75I (data not shown), which explains why this mutant was unresponsive in luciferase assays.

To learn how the M75L variant is constitutively active, we performed HDX-MS to probe structural and dynamical changes in the mutant at fast deuterium exchange time scales (t=1–60 min). With peptide coverage ranging from 85% to 89% (*Figure 4A*) for the AncSR2 and M75L LBDs, we monitored the dynamics of nearly the entire protein at different time points (*Figure 4—figure supplements 1–4*). First, we identified the effects of the M75L mutation on WT AncSR2 dynamics using a comparative HDX (ΔHDX) analysis (*Figure 4B*). An increase in deuterium uptake was observed in multiple regions of the M75L mutant, including residues in and adjacent to the ligand binding pocket: H3, H10, H6, and H7. This deprotection may indicate general destabilization of these LBD regions.

However the localization of these changes to the binding pocket may also suggest that M75L allows the pocket to sample a range of conformations, including some that resemble ligand-bound states which may permit constitutive activity. Deprotection is also unexpectedly observed in distant regions such as H9 and the N-terminal end of H10.

To explore the dynamic effects of the M75L mutation on ligand binding, we analyzed ΔHDX comparing the progesterone-bound forms to their apo counterparts for both WT and M75L receptors (*Figure 4—figure supplement 2*). While modest changes in deuterium exchange accompanied progesterone binding in M75L (*Figure 4C*), dramatic enhancement in dynamics was observed across AncSR2, including peptides [20]AGYDNTQPDTTNYLL[34], [48]VVKWAKALPGFRNLHLDD[65], [104]NEQRMQQSAM[113], [145]LLSTVPKEGLKSQ[160], suggesting that the two variants are differentially affected by the addition of progesterone (*Figure 4D*). A similar effect was seen in the estradiol-bound forms of the receptors (**Figure S5**). WT AncSR2 showed deprotection in several regions when bound to estradiol, while M75L showed no changes. However, a slight deprotection was observed which may highlight differences in M75L binding to estrogens vs 3-ketosteroids (*Figure 4—figure supplement 2*). These experimental observations strongly support a model in which the M75L mutation shifts the ensemble conformation to a ligand-bound state, allowing the receptor to be less dynamically responsive to the addition of ligand. This effect may also explain the reduced ligand binding ability observed in the M75L variant.

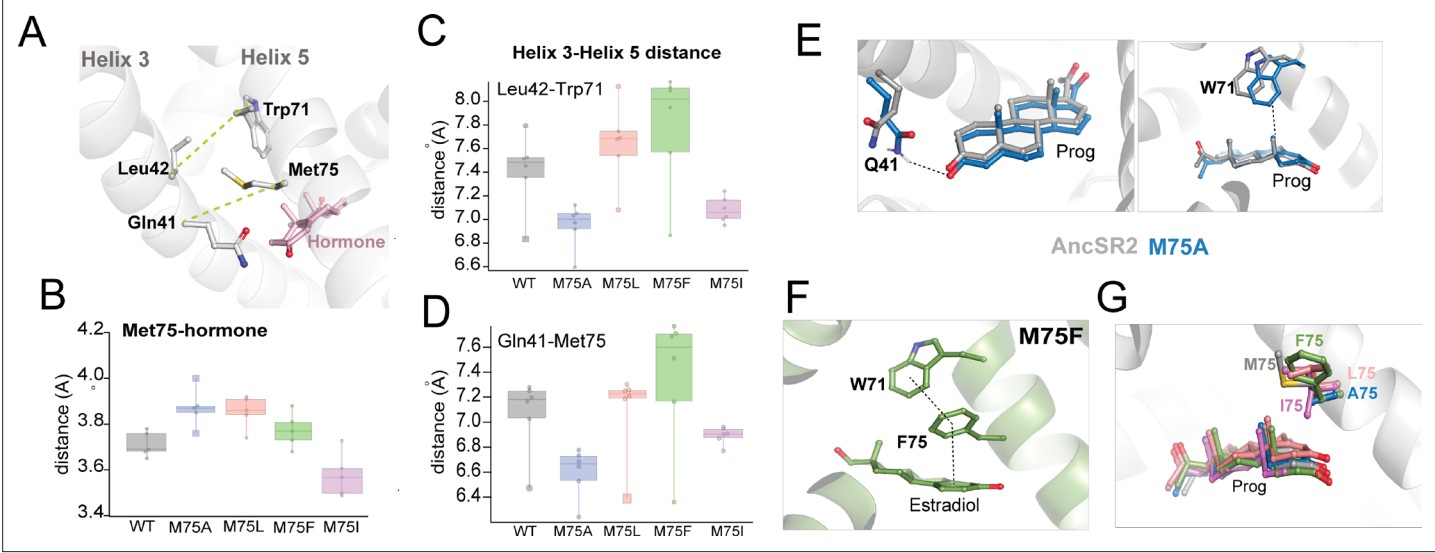

**Figure 5.** Contact measurements and conformational analysis of classical MD simulations. (**A**) AncSR2 structure indicating positions of M75 along with bound hormone and additional H5 and H3 residues used to determine contact measurements. To assess H3-H5 interhelical distances, measurements were performed between Leu42-Trp71 and Gln41-res75. (**B**) Average distances between residue 75 (**H5**) and A-ring of hormones. (**C**) Average distances between L42 Cα and Trp71 Cα atoms. M75A showed the smallest distances, M75F the largest, with all other complexes in between, indicating that the size of the sidechain determines the interhelical distance. (**D**) Average distances between Q41 Cα and res75 Cα atoms. This contact follows the same trends observed in (**C**). Individual data points in B, C, D represent distance measurements averaged over simulations. Each box and whisker representation displays the distribution of calculated distances for the five hormone-bound complexes and apo receptor. The lower bounds for each box in (**C** and **D**) correspond to the apo receptors. (**E**) Two potential explanations for enhanced activity in M75A are the proximity of Q41 (top) and Trp71 (bottom) to hormones, due to the reduced H3-H5 distances. (**F**) In estradiol-bound M75F, a pi-stacking triad is formed between Trp71, Phe75 and the hormone A-ring. This observation suggests an explanation for why the mutant is selectively activated by 3-ketosteroids but not by estrogens. (**G**) Simulations predict that the Ile sidechain of M75I is likely to insert into the binding pocket and interfere with ligand binding. This effect is not observed any of the other complexes.

The online version of this article includes the following figure supplement(s) for figure 5:

**Figure supplement 1.** Root mean square deviation (RMSD) calculated for all complexes from classical MD simulations.

**Figure supplement 2.** Distribution of calculated distances between Q41-ligand (left) and W71-ligand.

**Figure supplement 3.** M75A mutant receptor activation response abolished upon substitution of Q41A in the M75A background mutant (M75A-Q41A).

**Figure supplement 4.** Distances and angles between F75 and W71 are monitored for MD simulations of M75F complexes.

## MD simulations reveal conformational effects of M75 mutations

A key goal of this study is to determine the extent to which conformational effects predicted by MD simulations describe outcomes from molecular experiments. From our previous work on AncSR2 (*Okafor et al., 2020*), it is known that contact between H3-H5, as well as interactions between M75 and hormones may predict how well a hormone can activate AncSR2 (*Okafor et al., 2020*). To reveal the impact of M75 mutations on residue and ligand contacts, we used classical MD simulations to model each variant in the presence of the five hormones. We included an unliganded (apo) simulation for each variant, generating a total of thirty complexes. Root mean square deviation (RMSD) analyses of trajectories fluctuated around 2 Å or lower, confirming stability of the complexes over the length of the simulation (*Figure 5—figure supplement 1*). To measure contacts between two residues, we determined the minimum distance between heavy atoms (See Materials and methods) of both residues across the simulations. First, we measured the distance between residue 75 and the hormone (*Figure 5A*). In all mutants and for all ligands, residue 75 is within 4.5 Å of the hormone which is within the threshold for a van der Waals contact (*Figure 5B*). Of all variants, M75I has the shortest distances for this contact, with values ranging from 3.5 to 3.7 Å. Distances in all other variants are higher, ranging from 3.7 to 4.0 Å.

Next, we computed the H3-H5 interhelical distance in all variants by measuring Cα-Cα distances between two H3/H5 pairs: Gln41/res75 and Leu42/Trp71 (*Figure 5C and D*). Importantly, while the Ala sidechain is not able to form van der Waals contacts with any H3 residues, all other position 75 substitutes were bulky enough for contact (Data not shown). M75A and M75I complexes show the smallest interhelical distances, significantly shorter than other variants (*Figure 5D*). In several complexes, M75F shows the largest H3-H5 distances (as high as 8.1 Å between Leu42-Trp71) while M75L and WT AncSR2 show intermediate distances on average. Thus, we determined that while M75 mutations preserve the contact with hormones, they modulate the H3-H5 interhelical distance which may be a factor in the varying transcriptional responses observed here.

We then sought to visualize the conformational effects that accompany M75A, M75F and M75I substitutions in MD simulations. By performing a close examination of our MD trajectories of the M75A variant, we observed two new interactions that potentially stabilize hormones, formed by Trp71 (H5) and Gln41 (H3) (*Figure 5E*). Both residues are conserved in SRs: Trp71 mediates interactions between bound ligands and H12 (*Okafor et al., 2019*; *Duan et al., 2016*; *La Sala et al., 2021*) while Gln41 is positioned to stabilize the A-ring of hormones. Simulations predict that reduced bulk at position 75 allows both Trp71 and Gln41 sidechains to gain proximity to the hormone and potentially provide increased stability. We quantified this interaction using a distance analysis, showing that both residues are closer to the ligand in M75A complexes compared to WT (*Figure 5—figure supplement 2*). In support of this hypothesis, a Q41A mutation in the background of M75A abolished activation by 3-ketosteroids (*Figure 5—figure supplement 3*).

In M75F complexes, in addition to having the largest interhelical distances, (*Figure 5C and D*). we observe that Phe75 engages Trp71 in a hydrophobic interaction. We measured the fraction of time that the aromatic sidechains engage in pi-stacking and interestingly, the occupancy was less than 9% for 3-ketosteroid complexes but rose to ~40% in the M75F-estradiol complex (*Figure 5—figure supplement 4*). Additionally, Trp71, Phe75 and the aromatic A-ring of estradiol form a triad in this complex (*Figure 5F*), which was absent in 3-ketosteroid complexes. Thus, M75F bears strong similarities with AncSR2, where the Phe sidechain, similar to the WT Met, engages in pi interactions with estrogens but not 3-ketosteroids (*Okafor et al., 2020*). Unsurprisingly, while potency is reduced by 1–2 orders of magnitude, M75F displays a similar activation profile to WT AncSR2 (*Figure 2A and C*). Notably, DHT does not activate M75F in our assay, which we believe results from the same reason that DHT weakly activates AncSR2, that is, the lack of a C17 acetyl substituent to stabilize the D-ring end of the hormone via hydrogen bonding. Because the main difference between WT and M75F mutants is the increased H3-H5 distance resulting from the bulky Phe substitution, the larger distance may be responsible for the suboptimal functional profile of this mutant, a hypothesis that would require testing in future studies. Simulations of the M75I variant with ligands showed that the β-branched Ile sidechain is positioned to enter the binding pocket and interfere with ligand binding (*Figure 5G*), providing an explanation for the lack of ligand binding and transcriptional activation observed (*Figure 2E*).

## Clustering demonstrates that ligands selectively shift conformational states in AncSR2 variants

We previously reported that that NR ensembles generated by MD simulations experience conformational shifts upon addition of ligand that may reflect the activation potency of ligands (*Okafor et al., 2020*). To characterize these ligand-induced effects in our engineered receptors, we used accelerated MD simulations to achieve enhanced sampling of the conformational space for each receptor-hormone combination. By lowering the energetic barrier for conformational transitions during simulations, this method allows us to visualize conformational states that may not be sampled in classical MD. We obtained 500 ns accelerated MD trajectories in the apo state for all five variants along with ligand-bound states (aldosterone, progesterone, estradiol, cortisol, DHT) for AncSR2, M75F, M75A, and M75L. Excluding the binding incompetent M75I variant, accelerated MD was performed on 25 complexes in total.

To compare how the M75 mutations alter AncSR2 conformations, we performed combined clustering using frames (i.e. snapshots) obtained from the apo-AncSR2 trajectory with frames obtained from each of the apo-M75 mutants (*Figure 6A*). Each cluster is stacked, representing the fraction of the cluster comprised either of WT AncSR2 frames or frames from M75 mutants. M75A shows substantial conformational overlap with WT AncSR2, as all clusters containing M75A snapshots also contained large numbers of WT frames. This result suggests that the M75A mutation does not have a huge impact on the overall conformational state of WT AncSR2. We observe the opposite trend when comparing M75F to WT: both complexes cluster into two unique (non-overlapping) clusters, suggesting that this mutation induces a large conformational effect in AncSR2. M75I and M75L reveal very similar patterns to one another: while the mutant receptors are largely retained in one cluster, the wildtype complexes segregate into 4–5 smaller clusters (*Figure 6A*). This result suggests that a minor conformational state from WT AncSR2 is stabilized by these mutations.

Next, we used combined clustering to determine how ligand binding alters conformations within each variant (*Figure 6B–E*). We co-clustered frames from cortisol, DHT, aldosterone, progesterone, and estradiol-bound trajectories with frames from the corresponding apo simulation. Fractional populations comprising each cluster are provided in *Supplementary file 2*, while each stacked cluster in *Figure 6B–E* represents the fraction comprised of either apo or ligand-bound frames. In M75A, progesterone and aldosterone complexes form non-overlapping clusters with apo-M75A, while estradiol, DHT and cortisol-M75A show varying degrees of overlap, that is shifts in the conformational ensemble (*Figure 6B*). In contrast to M75A, all liganded M75F complexes show overlap with apo-M75F, suggesting that addition of ligand does not induce a large shift in the M75F conformational ensemble (*Figure 6C*).

In M75L, all ligands cause a shift in the conformational ensemble resulting in non-overlapping ligand-bound and apo clusters (*Figure 6D*). We also observe an interesting pattern where the apo complex always forms two large clusters with very similar compositions, regardless of the ligand it is co-clustered with. This pattern may suggest that a ligand-independent conformational response is induced in all M75L complexes. Follow up studies would be required to test this idea. For comparison, we also performed clustering on previously obtained accelerated MD trajectories of WT AncSR2 (*Okafor, 2021*). While the numbers and compositions of clusters change due to use of a different clustering radius in this work, the previous patterns remain the same: that is, 3-ketosteroid ligands tend to eliminate conformational overlap with apo AncSR2 while estradiol yields clusters largely overlapping with apo AncSR2 (*Figure 6E*).

As a visual inspection of clustering results predicts a relationship between ligand activity and overlap in clustering, we quantified the 'overlap fraction' for each receptor-ligand complex, that is, the fraction of apo frames that emerge in the same cluster with ligand-bound frames (see Methods). Overlap fraction values range from 0 (no overlap, e.g., M75A-aldosterone) to 1 (complete overlap, e.g., M75F-aldosterone). We observe a correlation between the overlap fraction and ligand activity measured by (i) $EC_{50}$ (*Figure 6F*), (ii) fractional fold activation at 10 nM and (iii) 100 nM (*Figure 6G and H*). Thus, results suggest that the ability of ligands to shift conformational ensembles of AncSR2 and variants may predict their in vitro transcription activity.

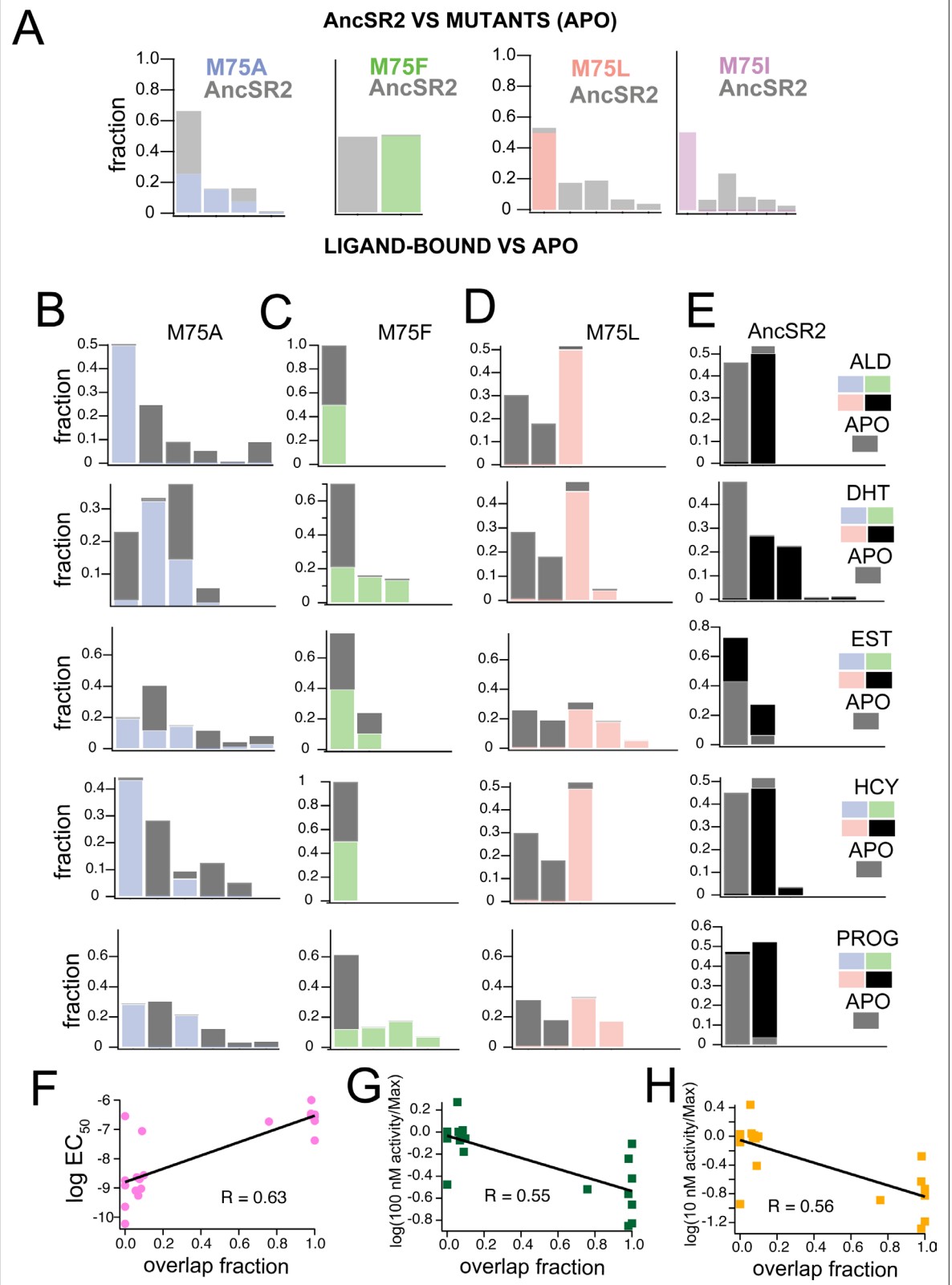

**Figure 6.** Clustering of accelerated MD simulations and correlation analysis. (**A**) Conformations obtained from accelerated MD simulations of unliganded AncSR2 were co-clustered with frames obtained from M75 mutants. M75A reveals substantial conformational overlap with AncSR2, M75F reveals no AncSR2 overlap while both M75L and M75I stabilize a minor conformational substate of AncSR2. Each stacked cluster represented the population obtained from WT AncSR2 frames (gray) stacked on the fraction obtained from mutant complexes (blue/green/pink/purple). (**B–E**) For each

*Figure 6 continued on next page*

Figure 6 continued

AncSR2 variant (except M75I), frames from ligand-bound simulations were co-clustered with conformations from the unliganded simulation of the same receptor. In M75A (**B**), M75L(**D**) and WT AncSR2 (**E**) all hormones introduce new conformational states to varying extents. In M75F, all liganded complexes show overlap with the apo receptor. Each stacked cluster represented the population obtained from apo frames (gray) stacked with the fraction of the cluster containing ligand-bound frames (blue/green/pink/black). (**F–H**) Overlap fraction shows correlation with EC$_{50}$ (**F**), and fractional fold activation, measured as fold change at 100 nM (**G**) and 10 nM (**H**) ligand divided by E$_{max}$/efficacy.

## Conformational analysis of MD-generated ensembles

Finally, we sought to reveal the conformational features defining the ensembles generated from M75 mutants. To quickly identify the most distinguishing features of each cluster from *Figure 6B–E*, we calculated root mean square fluctuations (RMSF) for all cluster populations (*Figure 7—figure supplement 1*). While these reveal that flexible regions of AncSR2 variants largely drive conformational differences between clusters, they also identify residues 100–125, that is H6 and H7 as a region that distinguishes clusters for all complexes, most prominently in WT AncSR2 complexes (*Figure 7—figure supplement 1*). Next, we performed more specific analyses on unliganded M75F, M75I, and M75L, as these three variants revealed a shift in the conformational ensemble when clustered with WT AncSR2 (*Figure 6A*). To achieve this conformational analysis, we obtained 100 representative

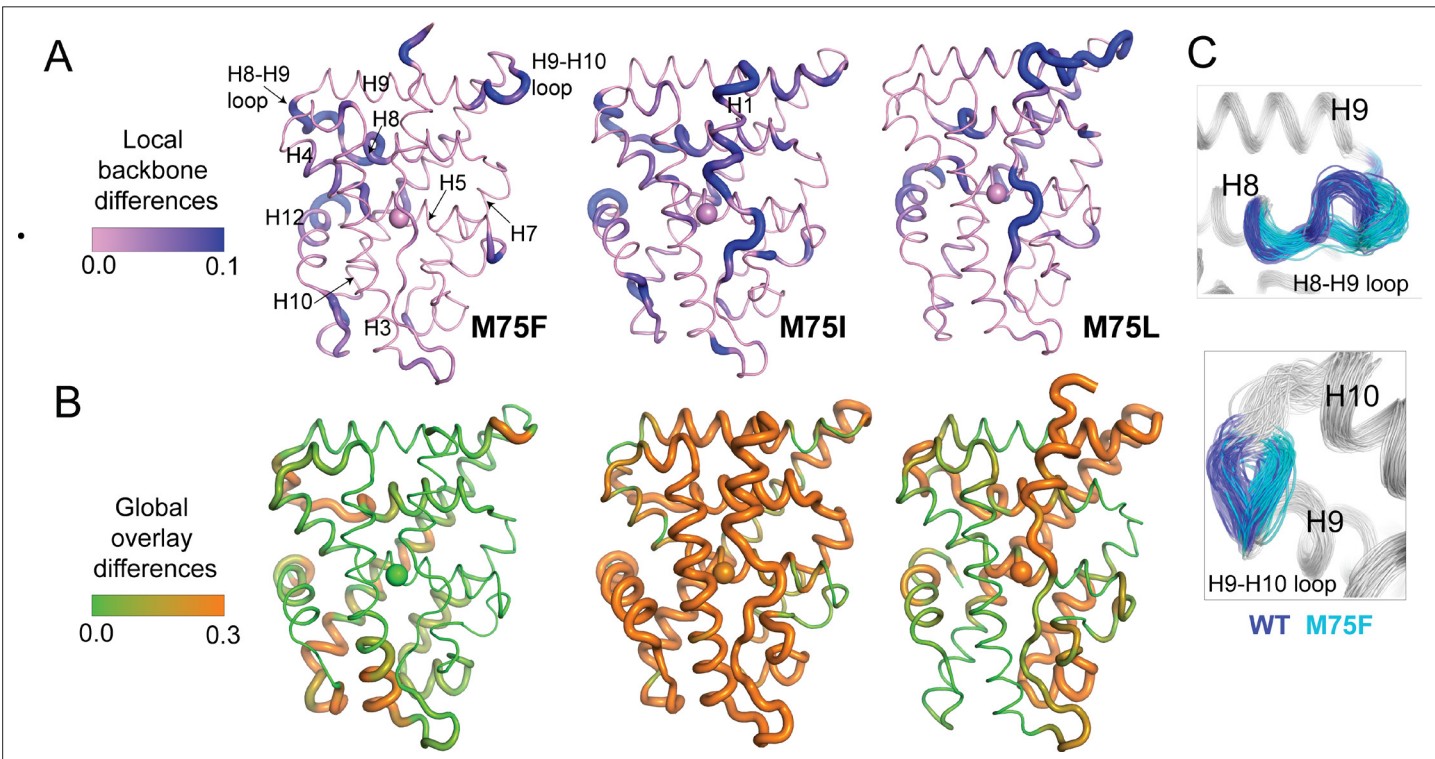

**Figure 7.** Structural analysis of MD simulations of AncSR2 variants to explain altered transcriptional response. (**A**) Ensemblator analysis of local/backbone changes of M75 variants compared to WT AncSR2. Conformational changes are quantified and colored by the discrimination index (DI). Lower DI values indicate regions where structures are nearly identical between WT and mutant receptors while higher DI indicates changes in local backbone angles induced by mutations. Spheres on H5 identify position 75. (**B**) Ensemblator analysis of global changes in M75 mutants compared to WT. Structures are globally overlaid prior to analysis. Structures are colored by DI, where lower values correspond to highly similar regions in superimposed structures while higher values indicate regions of structural dissimilarity. Spheres on H5 identify position 75. (**C**) Overlay of 200 structures used for Ensemblator analysis showing conformational changes observed in H8-H9 and H9-H10 loops. Structures from WT AncSR2 are colored blue while mutant structures are shown in cyan, illustrating that structural variations are the result of mutation.

The online version of this article includes the following figure supplement(s) for figure 7:

**Figure supplement 1.** Root mean square fluctuations (RMSF) for each cluster, by receptor-ligand complex.

**Figure supplement 2.** Discrimination Scores reporting on the global and local conformational differences between ensembles of WT AncSR2 and mutants.

structures each from the largest WT and mutant clusters of *Figure 6A*. We compared the subgroups using Ensemblator (*Brereton and Karplus, 2018*; *Clark et al., 2015*) to quantify both differences and similarities between mutant and WT populations. This method achieves local conformational comparison by calculating the similarity between backbone conformations based on dihedral angles (*Clark et al., 2015*). Global comparisons are performed based on an atom-level overlay of all 200 structures. Comparisons are represented by Discrimination Index (DI), a metric ranging between 0 and 1 that reveals the most significant local and global differences between both subgroups (See Materials and methods).

Conformational changes are colored by calculated DI, where higher values identify structural differences in the mutant relative to WT (*Figure 7A and B*). Here, we observe small changes in local (<0.3) and global DI (<0.5) for most regions of the receptors (*Figure 7—figure supplement 2*), indicating that subtle conformational changes drive the structural effects detected via clustering. Thus, we have focused on analyzing these subtle differences between mutant and WT AncR2 structures. In all variants, M75 mutation induces a backbone change in H5 beginning at Met72 (M75I) or Trp71 (M75F, M75L). These changes propagate to the adjacent H8, subsequently influencing the H8-H9 loop and/or the H9-H10 loop (*Figure 7C*). Other regions impacted include H10 and the pre-H12 loop. Global changes vary more drastically between the three variants. M75F shows the largest effects in H8, H10 and the bottom of H3. Conversely, M75I undergoes global shifts >0.3 DI in nearly all helices, while M75L is most affected at H6, H7, H8, and H9/H10. Interestingly, we note that some of the large global changes in M75L (H6, H7, H9) coincide with regions predicted by HDX-MS to be destabilized by the M75L mutation (*Figure 4B*).

## Discussion

Here, we have used molecular dynamics simulations to generate conformational ensembles of NR complexes, with a goal of revealing how ligand-induced population shifts correlate with the functional properties of ligands. We engineered AncSR2 variants with altered potency and specificity by substituting M75, a known regulator of activation and ligand recognition in steroid receptors. Our results reveal a strong relationship between ligand potency and the extent of population shifts in ligand-bound ensembles relative to apo NR ensembles. Clustering shows that active ligand complexes are unlikely to overlap with unliganded complexes while inactive ligands tend to show high to complete conformational overlap, suggesting that the addition of ligand does not shift the receptor out of a ligand-free state. We observe complexes yielding partial overlap in clustering, which implies a partial switch of the conformational ensemble. Future studies will be important to determine whether MD-generated ensembles of NR ligand complexes can be closely parsed to distinguish between partial agonist (low-efficacy) vs weak agonists (low-potency). Notably, because the M75 mutations do not perturb the AF-2 surface, the ensembles reflect extremely subtle conformational changes originating at H5 and thus highlight the sensitivity of this computational assay.

By combining biophysical experiments with a structural analysis of our MD trajectories, we determined the mechanisms underlying the altered functional profile of each NR variant. While we anticipated a loss of function with the M75A mutant due to the elimination of interhelical H3-H5 contacts, we observed that the variant retains activity similar to the WT receptor. An explanation that emerges from our MD simulations is that one or both of two conserved pocket residues, Trp71 and Gln41 may gain proximity to the hormone and provide stabilization. Another contributing factor, based on previous SR studies is that the lack of a bulky sidechain at position 75 may create excess volume in the binding pocket, allowing ligands to bind without perturbing H12 or the AF-2. Such a state would explain the constitutive activity and/or gain of function observed in M75A mutants of PR and MR (*Zhang et al., 2005*).

Conversely, the larger Phe substitution introduces the largest H3-H5 distances in our simulations (*Figure 5C and D*). As there are no other structural changes observed across the variant, we hypothesize that the reduced potency and the lower thermodynamic stability in M75F may result from the increased interhelical distance. However, simulations reveal that the Phe sidechain distinguishes between aromatized and non-aromatized hormones via pi interactions, confirming the role of position 75 in mediating ligand specificity. These pi interactions with estradiol cause frustration around the A-ring, preventing stabilization and activation of WT AncSR2 and M75F by estrogens. Additionally, our results support the previously established importance of steroid D-ring interactions with the

conserved H3 residue, N37 (*Rafestin-Oblin et al., 2002*; *Dezitter et al., 2014*; *Petit-Topin et al., 2009*). Because neither estradiol nor DHT activate M75F and both lack the ability to interact with N37, this residue could be playing a compensatory role in this impaired variant, permitting activation by aldosterone, hydrocortisone, and progesterone.

The most unexpected result was the constitutive activity observed in the M75L mutant, surprising because the variant retained similar stability to the WT receptor with only slight differences in secondary structure (*Figure 1C and D*). Ligand binding assays showed that M75L has reduced affinity for our 11-DOC-FAM probe compared to WT AncSR2, as well as a lower measured $K_i$. Increased conformational dynamics around the ligand binding pocket in the apo M75L receptor compared to WT in HDX-MS studies suggests that the mutation likely affects ligand binding, supported by reduced $K_i$s observed in binding assays. However, the absence of a change in ΔHDX upon addition of ligand to M75L supports our claim that the receptor adopts a partially-active state that mimics ligand-bound AncSR2, resulting in low levels of constitutive activity and a reduced response to ligands.

This work demonstrates that an in silico approach can be used for describing ligand-specific conformational changes in NR ensembles. Because of the demonstrated importance of NR ensembles for understanding and predicting activity profiles of ligands, our findings confirm the inherent promise in the use of MD-generated ensembles as a predictive tool in ligand design. An added advantage is that analysis of MD trajectories is useful for providing a structural description of the ensemble, as well as elucidating the mechanisms by which ligands induce distinct conformational effects in NRs. We have used Ensemblator to perform reveal the local and global structural perturbations associated with three of the mutations investigated. We also observe that M75 mutations induce dynamic effects at distant regions of the receptor, including helices 9 and 10, which will be explored in future work to determine the potential for functional modulation of receptors via novel mechanisms.

# Materials and methods

**Key resources table**

| Reagent type (species) or resource | Designation | Source or reference | Identifiers | Additional information |
|---|---|---|---|---|
| Strain, strain background (*Escherichia coli*) | BL21(DE3) | New England Biolabs | C2527 | Chemically competent cells |
| Strain, strain background (*Escherichia coli*) | DH5α | New England Biolabs | C2987 | Chemically competent cells |
| Cell line (*Homo-sapiens*) | Hela | ATCC | - | Cell lines maintained in Penn State Sartorius Cell Culture facility |
| Cell line (*Homo-sapiens*) | CHO | ATCC | - | Cell lines maintained in Penn State Sartorius Cell Culture facility |
| Antibody | Anti-Gal4-DBD (Mouse monoclonal): (RK5C1) | Santacruz Biotechnology | sc-510 | WB (1:200) |
| Antibody | m-IgG Fc BP-HRP (Mouse polyclonal) | Santacruz Biotechnology | sc-525409 | WB (1:1000) |
| Antibody | Anti-actin (rabbit polyclonal) | Sigma | A2066 | WB (1:8000) |
| Antibody | Goat-anti Rabbit IgG H+L (HRP) (Goat polyclonal) | Abcam | Ab97051 | WB (1:5000) |
| Recombinant DNA reagent | pSG5-Gal4-DBD fused LBD (plasmid) | Reference *Eick et al., 2012* | - | *Eick et al., 2012*, Plos Genetics |
| Sequence-based reagent | M75L_F | IDT | PCR primers | 5'- CTGGATGGGCCTGTTGGCCTTCGCCAT-3' |
| Sequence-based reagent | M75L_R | IDT | PCR primers | 5'- ATGGCGAAGGCCAACAGGCCCATCCAG –3' |

*Continued on next page*

Continued

| Reagent type (species) or resource | Designation | Source or reference | Identifiers | Additional information |
|---|---|---|---|---|
| Sequence-based reagent | M75A_F | IDT | PCR primers | 5'-CCTGGATGGGCCTGGCGGCCTTCGCCATGG-3' |
| Sequence-based reagent | M75A_R | IDT | PCR primers | 5'-CCATGGCGAAGGCCGCCAGGCCCATCCAGG-3' |
| Sequence-based reagent | M75F_F | IDT | PCR primers | 5'- CTGGATGGGCCTGTTCGCCTT CGCCATGG-3' |
| Sequence-based reagent | M75F_R | IDT | PCR primers | 5'-CCATGGCGAAGGCG AACAGGCCCATCCAG-3' |
| Sequence-based reagent | M75I_F | IDT | PCR primers | 5'-CCTGGATGGGC CTGATAGCCTTCGCCATG-3' |
| Sequence-based reagent | M75I_R | IDT | PCR primers | 5'-CATGGCGAAGGCTATCAGGCC CATCCAGG-3' |
| Sequence-based reagent | M75A-Q41A_F | IDT | PCR primers | 5'-GGCTGGCCGAGA AGGCGCTGGTGTCTGTGG-3' |
| Sequence-based reagent | M75A-Q41A_R | IDT | PCR primers | 5'-CCACAGACACCAG CGCCTTCTCGGCCAGCC-3' |
| Commercial assay or kit | Nucleospin Plasmid purification kit | Clontech | Clontech:639647 | Used for Plasmid DNA purification according to manufacturer's protocol. |
| Commercial assay or kit | Dual-Glo Luciferase kit | Promega | E2980 | - |
| Chemical compound | FuGENE HD | Promega | E2311 | - |
| chemical compound, drug | β-Estradiol | Sigma Aldrich | E8875 | - |
| chemical compound, drug | Progesterone | Acros Organics | AC225650050 | - |
| chemical compound, drug | Aldosterone | CAYMAN Chemical | 15273 | - |
| Chemical compound, drug | Hydrocortisone 98% 1GR | Acros Organics | AC352450010. 103515–190 | - |
| Chemical compound, drug | 11-Deoxycorticosterone Acetate 97% | Acros Organics | 460470010 | - |
| Chemical compound, drug | Dihydrotestosterone(DHT) | Selleckchem.com | S4757 | - |
| Chemical compound, drug | Dimethyl Sulfoxide (DMSO) | VWR | BDH115-1LP | - |
| Chemical compound, drug | Glycine 99+%, Molecularbiology grade, Ultrapure | Thermo Fisher Scientific | J16407-A1 | - |
| Chemical compound, drug | Ammonium Persulfate 98+% ACS reagent | Millipore Sigma | 248614–500 G | - |

*Continued on next page*

*Continued*

| Reagent type (species) or resource | Designation | Source or reference | Identifiers | Additional information |
|---|---|---|---|---|
| Chemical compound, drug | Glycerol Certified ACS | Fisher Scientific | G33-4 | - |
| Chemical compound, drug | DL-Dithotheitol | Sigma-Aldrich | D0632-5G | - |
| Chemical compound, drug | DL-Dithotheitol | Alfa Aesar | A15797 | - |
| Commercial assay or kit | Quick Start Bradford 1 X | Bio-Rad | 5000205 | Used according to manufacturer's protocol |
| Commercial assay or kit | Pierce ECL Western Blotting substrate | ThermoFisher Scientific | 32109 | Used according to manufacturer's protocol |
| Software, algorithm | Prism 9 | Graph Pad Prism | GPS-1988381 | - |
| Software, algorithm | AMBER 2020 | *Case, 2020* | https://ambermd.org | |
| Software, algorithm | Carma | *Glykos, 2006* | https://utopia.duth.gr/ glykos/Carma.html | |
| Software, algorithm | MMTSB | *Feig et al., 2004* | http://blue11.bch.msu.edu/mmtsb/Main_Page | |
| Software, algorithm | VMD | *Humphrey et al., 1996* | RRID: SCR_001820 | |
| Other | Yeast Extract Powder | RPI | Y200250 | Used for preparation of Luria Bertini Media. See in Methods subsection section's "**Cloning, expression, and purification of ligand binding domain of WT and mutants**" |
| Other | Tryptone | RPI | T60060 | Used for preparation of Luria Bertini Media. See in Methods subsection section's "**Cloning, expression, and purification of ligand binding domain of WT and mutants**" |
| Other | Trypsin | Corning | 25–053 C1 | See Methods subsection Western blotting and Luciferase reporter assay |
| Other | Fetal Bovine Serum | Corning | 35-072CV | See Methods subsection Western blotting and Luciferase reporter assay |
| Other | Phosphate- Buffered Saline | Corning | 21–040-CV | Used for washing of Hela and CHO cells during cell passage. See Methods subsection Western blotting and Luciferase reporter assay |
| Other | NaCl | RPI | Y200250 | Used for preparation of Luria Bertini Media and different buffers. See in Methods subsection section's "**Cloning, expression, and purification of ligand binding domain of WT and mutants**" |
| Other | Ethidium Bromide solution | VWR | 97064–970 | Used to visualize electrophoresed DNA. See Methods subsection "**Cloning, expression, and purification of ligand binding domain of WT and mutants**" |
| Other | Tris-Base | Fisher Scientific | BP152-500 | For preparation of protein purification buffers. See in Methods subsection section's "**Cloning, expression, and purification of ligand binding domain of WT and mutants**" |

*Continued on next page*

*Continued*

| Reagent type (species) or resource | Designation | Source or reference | Identifiers | Additional information |
|---|---|---|---|---|
| Other | HEPES | Fisher Scientific | BP310-500 | Used in ligand binding assay. See Methods subsections "Cloning, expression, and purification of ligand binding domain of WT and mutants" and ligand binding assay |
| Other | Sodium Deoxycholate | Sigma | D6750 | See Methods subsection Western blotting |
| Other | 3X-Gel loading dye | New England Biolabs | B7703 | For protein sample load during SDS-PAGE (*Figure 1—figure supplement 1*). See Methods subsection "Cloning, expression, and purification of ligand binding domain of WT and mutants" |
| Other | 6X-Gel loading dye | New England Biolabs | B7025S | For DNA loading on agarose gels. See Methods subsection "Cloning, expression, and purification of ligand binding domain of WT and mutants" |
| Other | Pre-stained Protein Ladder | New England Biolabs | P04772S | See Methods subsection "Cloning, expression, and purification of ligand binding domain of WT and mutants" |
| Other | Every Blot Blocking Buffer | Bio-Rad | 12010020 | Used according to manufacturer's protocol. See Methods subsection Western blotting. |
| Other | Syringe filter (0.22 micron) | Fisherbrand | 09-720-511 | See Methods subsection "Cloning, expression, and purification of ligand binding domain of WT and mutants." |
| Other | Halt Protease & Phosphatase Inhibitor | ThermoFisher Scientific | 78440 | Used according to manufacturer's protocol. See Methods subsection Western blotting |
| Other | Amicon-Ultra-15 | Millipore, USA | UFC901024 | See in Methods subsection section's "Cloning, expression, and purification of ligand binding domain of WT and mutants" |
| Other | EcoRI-HF | New England Biolabs | R3101S | Used according to manufacturer's protocol. See in Methods subsection "Cloning, expression, and purification of ligand binding domain of WT and mutants" |
| Other | HindIII-HF | New England Biolabs | R3104S | Used according to manufacturer's protocol. See in Methods subsection "Cloning, expression, and purification of ligand binding domain of WT and mutants" |
| Other | DpnI | New England Biolabs | R0176S | Used according to manufacturer's protocol for methylated DNA stand degradation. See Methods subsection "Cloning, expression, and purification of ligand binding domain of WT and mutants" |
| Other | Nuvia-IMAC | Bio-Rad | 780–0812 | See Methods subsection "Cloning, expression, and purification of ligand binding domain of WT and mutants" |
| Other | ENrich SEC70 | Bio-Rad | 7801070 | See Methods subsection "Cloning, expression, and purification of ligand binding domain of WT and mutants" |
| Other | ENrich SEC650 | Bio-Rad | 7801650 | See Methods subsection "Cloning, expression, and purification of ligand binding domain of WT and mutants" |

## Materials

Sodium chloride, Tris base, glycine, sodium dodecyl sulphate, ethylenediaminetetraacetic acid (EDTA), imidazole, glycerol were purchased from Fischer scientific (USA). Ampicillin, tryptone, yeast extract, isopropyl β-D-1 thiogalactopyranoside (IPTG) were procured from RPI chemicals (USA). DTT was purchased from Alfa Aesar/Sigma, USA. Estradiol and ammonium persulphate were purchased from Sigma Chemical Co. (USA). Hydrocortisone, 11-Deoxycorticosterone (11-DOC), progesterone,, 11-Deoxycyortisterone acetate were purchased from Acros organics. Dihydrotestosterone and aldosterone purchased from Selleckchem, USA and Cayman Chemicals, respectively. Bioscale Nuvia-IMAC

Ni-charged and ENrich SEC70 and SEC650 10/300 size exclusion columns were purchased from Biorad, USA and used with BioRad NGC Quest plus FPLC system. Syringe filters (0.2 micron) were procured from Millipore corporation. All reagents and chemicals were of analytical grade. HeLa and CHO cells have been used, obtained from ATCC. Mycoplasma testing confirmed no contamination.

## Cloning, expression, and purification of ligand binding domain of WT and mutants

The gene encoding AncSR2 was PCR amplified from the vector pSG5-Gal4-DBD-SR2-LBD using forward and reverse primers containing *Eco*RI and *Hind*III restriction sites, respectively, to clone into pMALCH10T vector. Primers were designed (*Supplementary file 1*), followed by mutagenesis to generate M75L, M75A, M75I, and M75F mutants of AncSR2-LBD in both pSG5 and pMALCH10T vectors. Mutants were confirmed by DNA sequencing.

MBP-His-tagged LBDs of WT-AncSR2 and its mutants were expressed in and purified from *E. coli*. BL21(DE3) as previously reported with slight modification (*Eick et al., 2012*). Briefly, cells containing the respective plasmids were grown in LB broth till O.D$_{600}$ reaches 0.6–0.8 at 37 °C. The protein expression was induced by addition of 0.3 mM IPTG and 50 µM progesterone and grown further at 30 °C for 4 hr. The cells were harvested by centrifugation at 8000 RPM for 10 min. The cells were lysed by sonication using a 10 s pulse-on and 30 s pulse-off cycle. Cell debris was removed by centrifugation of lysate at 15,000 RPM for 40 min. The cleared supernatant was purified by Ni-Affinity chromatography (Nuvia-IMAC). The purified protein was subjected to TEV protease treatment (0.5 mg/ml) overnight and simultaneously dialyzed against buffer containing 20 mM Tris (pH 7.4), 150 mM NaCl and 10% glycerol. The dialyzed lysate was twice purified by a Ni-affinity column, followed by collection of flow through containing desired LBDs. LBDs were finally purified by SEC70 gel filtration column using Bio-Rad NGC plus system in a buffer containing 20 mM Tris (pH 7.4), 150 mM NaCl and 10% glycerol and stored in small aliquots at –20 °C until further use. It should be noted that all steps from cell harvesting to gel filtration were either performed on ice or at 4 °C, unless stated otherwise. The purity of the purified protein was assessed by loading the sample on the 14% SDS-PAGE.

For ligand binding assays, MBP-tagged AncSR2 LBD and mutants were expressed and purified (without protease treatment) as described above for the LBD, except for the use of 0.4 mM IPTG and 50 µM 11-DOC acetate for induction of protein expression, followed by overnight growth at 18 °C. The MBP tagged protein was purified by Ni-affinity chromatography, followed by gel filtration in pH 7.4 buffer containing 20 mM Tris (pH 7.4),150 mM NaCl and 10% glycerol. The final purification was performed using a SEC650 gel filtration column in a buffer containing 20 mM HEPES (pH 7.4), 150 mM NaCl, 3 mM EDTA, 5 mM DTT and 0.005% Triton X-100. The purified protein concentrated using 10 KDa cutoff Amicon.

## Circular dichroism measurements

Far UV-CD measurements were done on a Jasco J-1500 spectrophotometer equipped with a temperature controller. The far-UV CD spectra of the WT and its mutants were measured in the wavelength range 195–250 nm. For spectral measurements,1 mm path length cuvette was used, with scan rate of 50 nm/s, 1 s response time and bandwidth of 1 nm and protein concentration used was 0.2 mg/ml. The CD instrument was continuously purged with N$_2$ gas at 5–8 lit/min flow rate and routinely calibrated with D-10-camphorsulfonic acid. Each spectrum was an average of 3 consecutive scans and corrected by subtraction of the buffer (10 mM phosphate, pH 7.4 and 100 mM NaCl). The raw CD data was converted into mean residue ellipticity at a wavelength, $[\theta]_\lambda$ (deg cm$^2$dmol$^{-1}$) by using the relation,

$$[\theta]_\lambda = M_0\theta_\lambda/10lc \tag{1}$$

where, $M_o$ is mean residue weight of the protein, $\theta_\lambda$ is the observed ellipticity in millidegrees at $\lambda$ wavelength, $c$ is the concentration of protein in mg ml$^{-1}$, and $l$ represents the cuvette path length in centimeters.

## Thermal denaturation measurements

Thermal denaturation was followed by measuring changes in the CD signal at 222 nm as a function of temperature. The heating rate of 1 °C with bandwidth 4 nm, 2 s response time was used in the temperature range 20–70°C to follow the denaturation. The raw CD data was converted into

the concentration independent parameter ($[\theta]_\lambda$) using equation 1. In the analysis of the denaturation curves, a two-state model (N=native state, D=denatured state) was assumed, and the temperature dependencies of pre- and post-denaturation baselines are linear. Stability curves ($\Delta G$ vs temperature) were constructed choosing values of $\Delta G$ (±1.3 kcal/mol) close to the midpoint of denaturation ($T_m$) that fall on the straight line. A linear least square analysis was used to estimate the entropy change at $T_m$ (=- $\delta\Delta G/\delta T)_p$ which is then multiplied by $T_m$ to get the values of apparent $\Delta H_m$. It should be noted that the heat-induced denaturation process was irreversible in the measured experimental conditions, so the stability parameter is defined here as apparent $T_m$.

The fraction of denatured molecules ($f_D$) was calculated by the relation:

$$f_D = (y y_N / y_D - y_N) \tag{2}$$

where y is the observed optical property of protein at temperature $T$, $y_N$ and $y_D$ are optical properties of native and denatured molecules at the same temperature.

## Ligand binding and competition assay

WT-AncSR2 and the M75L mutant were expressed and purified as MBP-tagged proteins. All fluorescence polarization experiments were performed buffer containing 20 mM HEPES (pH 7.4), 150 mM NaCl, 3 mM EDTA, 5 mM DTT and 0.005% Triton X-100. For saturation binding experiments, the binding affinity ($K_d$: dissociation constant) of the receptor for the probe dexamethasone-fluorescein (11-DOC-FAM) was determined using a constant concentration of 10 nM of the probe and a variable receptor protein concentration of $7.5\times10^{-6}$ – $4.5\times10^{-10}$ M in a 384 well plate. The plate was centrifuged at 500 RPM for 2 min and incubated overnight at 4 °C before reading. Fluorescence polarization measurements were performed on a Spectramax iD5 plate reader (Molecular Devices, USA) using excitation and emission wavelengths of 485 and 528, respectively. Six technical replicates and two biological replicates were obtained and data was plotted as the average mP (millipolarization) of all replicates versus receptor protein concentration. The saturation binding curve was analyzed by a one-site hyperbola binding model using GraphPad Prism vs 9 (GraphPad, Inc, La Jolla, USA). In the competition binding assay, 10 nM 11-DOC-FAM and protein concentration approximately 1.1–1.8 times $K_d$ for 11-DOC-FAM were incubated with variable $10^{-10}$–$10^{-5}$ M (competitive ligand overnight at 4 °C). The observed mP values in the presence of competitive ligands were plotted and fit by the *Fit Ki* model of GraphPad Prism vs 9. All data points were plotted after buffer subtraction.

## Luciferase reporter assays

Hela cells were grown and maintained in phenol red free medium MEM-α supplemented with 10% charcoal-dextran stripped FBS. Cells were seeded in 96-well plates at 70–90% confluency and co-transfected with 1 ng Renilla (pRL-SV40), 50 ng 9x-UAS firefly luciferase reporter and 5 ng pSG5-Gal4DBD-LBD fusions of WT-AncSR2, M75L, M75A, M75I, and M75F receptor plasmids using FuGene HD (Promega). Cells were treated with DMSO or varying drug concentrations 24 hr after transfection, all in triplicate. Firefly and Renilla luciferase activities were measured 24 hr after drug treatment using Dual-Glo kit (Promega) using a Spectramax iD5 plate reader. Fold activation is represented as normalized luciferase over DMSO control. Dose response curves were generated by GraphPad Prism v9.0.

## Western blotting

Hela cells maintained in MEMa and 10% FBS at 37 °C. For transfection, $0.8–1\times10^{-6}$ cells were seeded in six-well plate (Corning, USA). After 24 hr, 2 µg DNA of WT AncSR2 and mutant plasmids were transfected individually in different wells of the plate using FuGene HD transfection agent. Cells were then grown for 48 hr then harvested and lysed in a buffer (50 mM Tris, pH 8.0, 150 mM NaCl, 0.1% Triton X-100, 0.5% Sodium Deoxycholate and 1 mM Sodium azide and protease inhibitor (Thermofisher Scientific, USA) by incubating it on ice for 30 mins. The cell lysates were cleared by subjecting it to centrifugation at 12,000 rpm for 30 min at 4 °C. The sample concentration determined by comparing with standard plot of Bovine Serum Albumin. The protein concentration estimated by Bradford reagent (Bio-Rad, USA). An equal amount (40 µg) of total protein samples were electrophoresed on 12% SDS-PAGE and transferred to a PVDF membrane. Gal4-DBD fused protein was detected using ECL (Thermofisher Scientific, USA) after incubation of mouse monoclonal anti-GAL4DBD antibody (sc510, Santacruz Biotechnology, Santa cruz, USA)) and horse reddish peroxidase linked secondary antibody

(sc-525409, Santacruz Biotechnology, Santa cruz, USA). The same blot stripped off and then restained with rabbit raised anti-actin polyclonal antibody (A2066, Millipore Sigma, USA) and detected by Goat raised anti-rabbit horse reddish peroxidase linked secondary antibody (ab97051, Abcam, USA).

## Molecular dynamics simulations

Coordinates for AncSR2 ligand binding domain were obtained from PDB 4FN9, AncSR2 LBD-progesterone (*Eick et al., 2012*). Ligand complexes with the other hormones in this study (dihydro-testosterone, hydrocortisone, aldosterone, estradiol) were constructed by manually modifying the steroidal core of progesterone. All waters and surface-bound molecules from the crystallization buffer were deleted from the models. M75 mutant AncSR2 complexes were prepared by using Xleap in AmberTools20 (*Case, 2020*) to replace the M75 sidechain with alanine, leucine, isoleucine and phenyl-alanine respectively. In total, 30 complexes (5 AncSR2 variants, 5 hormones and 1 unliganded state per variant) were prepared for simulation. Unliganded states were prepared by removing the ligand from co-crystal structures followed by additional simulations.

All complexes were prepared using Xleap. Parameters for steroid hormones were obtained using the Antechamber (*Wang, 2001*) and the Generalized Amber ForceField (*Götz et al., 2012*). The parm99-bsc0 (*Pérez et al., 2007*) forcefield was used for protein residues. Briefly, complexes were solvated in an octahedral box of TIP3P water (*Jorgensen et al., 1983*), allowing a 10 Å buffer around the protein. Na +and Cl- ions were added to achieve a final concentration of 150 mM. Minimization was performed in four steps. First, 500 kcal/mol.Å$^2$ restraints were placed on all solute atoms, and 5000 steps of steepest descent performed, followed by 5000 steps of conjugate gradient minimization. In the second step, this protocol was repeated with restraints reduced to 100 kcal/mol.Å$^2$. Restraints were then removed from protein atoms and retained on the hormone for a third minimization step, followed by a final unrestrained minimization for all atoms.

Complexes were heated from 0 to 300 K using a 100 ps run with constant volume periodic boundaries and 5 kcal/mol Å$^2$ restraints on all solute atoms. All simulations were performed using AMBER 2020 on GPUs (*Götz et al., 2012*; *Salomon-Ferrer et al., 2013*) Before production MD simulations, 10-ns simulations with 10 kcal/mol.Å$^2$ restraints on all solute atoms were obtained in the NPT ensemble. This was followed by a second 10-ns simulation with restraints reduced to 1 kcal/mol.Å$^2$. Finally, restraints were retained only on the ligand atoms for a third 10-ns equilibration step. Production trajectories were obtained on unrestrained complexes, each complex simulated for 500 ns in triplicate. All MD was performed with a 2 fs timestep and using the SHAKE algorithm (*Ryckaert et al., 1977*) to fix heavy atom hydrogen bonds. Simulations were performed with the NPT ensemble, a cutoff of 10 Å to evaluate long-range electrostatics and particle mesh Ewald and van der Waals forces.

## Accelerated MD

Accelerated MD was used as previously reported (*Okafor et al., 2020*; *Hamelberg et al., 2004*) to enhance conformational sampling for AncSR2 complexes. We apply a dual-boosting approach, selecting parameters for potential energy threshold (EP), dihedral energy threshold (ED), dihedral energy boost (aD) and total potential energy boost (aP) using published guidelines (*Fratev, 2015*). 500 ns accelerated MD simulations were performed for a total of 30 complexes (5 AncSR2 variants, 5 hormones, and 1 unliganded state per variant). All simulations were performed in AMBER 2020 and classical MD simulations were used to obtain Average dihedral energy (EavgD) and average total potential energy (EavgP).

aD: 0.2 * (EavgD +3.5 (kcal/mol *Nsr))
ED: EavgD +3.5 kcal/mol * Nsr
aP: 0.16 kcal/mol * Natom
EP: EavgP + (0.16 kcal/mol*Natom) (where Nsr = number of total solute residues, Natom = total number of atoms)

## Analysis

Structural averaging and analysis were performed with the CPPTRAJ module of AmberTools17 (*Roe and Cheatham, 2013*). The 'strip' and 'trajout' commands of CPPTRAJ were used to remove solvent atoms and obtain twenty-five thousand evenly spaced frames from each simulation for analysis. For each complex, triplicate runs were combined to yield seventy-five thousand frames for analysis. Two

residues were defined to be within in Van der Waals contact if the distance between a pair of heavy atoms from each residue was <4.5 Å. To identify contacts, the 'distance' command of CPPTRAJ was used to measure the distance between all pairs of heavy atoms (i.e. non-H atoms) on both residues. For each frame in the analyses, the shortest pair distance was recorded and these values averaged at the end to obtain the miminum distance between heavy atoms.

## Clustering and conformational analysis

The MMTSB toolset (*Feig et al., 2004*) was used to perform clustering of accelerated MD trajectories. clustering analyses. For each complex, 25,000 evenly spaced conformations were obtained from each 500 ns trajectory for clustering. A 2.4 Å cutoff was used for all systems, which results in distinct numbers/sizes of clusters for each complexes but allows for an unbiased analysis. Clustering was performed between mutant and WT variants, as well as between liganded and apo complexes.

To quantify the extent of the ligand-induced conformational change in accelerated MD, we defined the 'overlap fraction' as the fraction of apo frames that emerge in the same cluster with ligand-bound frames. For each cluster in a combined clustering analysis, a 5% cutoff is applied to determine whether the cluster is a mix of both 'apo' and 'liganded' frames: that is the size of the smaller component must be greater than or equal to 5% of the larger component. E.g. if cluster 1 has 293 'apo' frames and 14287 'liganded' frames, this cluster is not considered mixed, as 293<5% * 14287. If cluster 1 has 1554 'apo' frames and 3269 'liganded' frames, this cluster is considered mixed, as the 5% threshold is fulfilled. Next, the overlap fraction is calculated by summing over the total number of apo frames in all mixed clusters. This criteria ensures that all apo frames counted comprise a non-negligible population of the mixed cluster, identifying them as unliganded receptor conformations that are not eliminated (i.e. 'shifted') by addition of ligand. If no mixed clusters are present, the overlap fraction is 0, suggesting that ligand binding led to a 'complete' shift in the conformational ensemble. An overlap fraction of 1 is obtained if all clusters are mixed.

## Amide hydrogen deuterium exchange mass spectrometry

Wild type and M75L AncSR2 samples were stored in 20 mM Tris, 150 mM NaCl, pH 7.4. To assess allostery in response to ligand binding (estradiol and progesterone), 10 μM SR2 samples were incubated with 200 μM ligand at 37 °C for 150 min before HDX. Deuterium labelling was carried out using a PAL RTC autosampler (LEAP technologies). All samples were diluted to a final concentration of 90.9% $D_2O$ to initiate the deuterium exchange reaction. Deuterium buffers were prepared by dilution of 20 X storage buffer in $D_2O$. Deuterium exchange was carried out at room temperature (20 °C) maintained on a drybath for 10, 30, 60, 900, and 3600 s followed by rapidly quenching the reaction to minimize back exchange using 1.5 M GdnHCl and 0.1% FA on ice to bring the pH down to 2.5.

Quenched samples were injected onto an immobilized pepsin treatment (BEH Pepsin Column, Enzymate, Waters, Milford, MA) using a nano-UPLC sample manager at a constant flow rate of 75 μl/min of 0.1% formic acid. Proteolyzed peptides were then trapped in a VanGuard column (ACQUITY BEH C18 VanGuard Pre-column, 1.7 μm, Waters, Milford, MA) and separated using a reversed phase liquid chromatography column (ACQUITY UPLC BEH C18 Column, 1.0×100 mm, 1.7 μm, Waters, Milford MA). NanoACQUITY binary solvent manager (Waters, Milford, MA) was used to pump an 8–40% acetonitrile gradient at pH 2.5 with 0.1% formic acid at a flow rate of 40 μl/min and analyzed on a SYNAPT XS mass spectrometer (Waters, Milford, MA) acquired in MS^E mode (*Lim et al., 2017*).

Undeuterated SR2 particles were sequenced by MS^E to identify pepsin digested peptides using Protein Lynx Global Server Software (PLGS v3.0) (Waters, Milford, MA). The peptides were identified by searching against the SR2 protein sequence database with a non-specific proteolysis enzyme selected. Peptides from the undeuterated samples that were identified and matched from the primary sequence database were filtered and considered with the following specifications: precursor ion tolerance of <10 ppm, products per amino acid of at least 0.2 and a minimum intensity of 1000.

Average deuterium exchange in each peptide was measured relative to undeuterated control peptides using DynamX v3.0 (Waters, Milford, MA) by determining the centroid mass of each isotopic envelope. Subtractions of these centroids for each peptide from the undeuterated centroid determined the average number of deuterons exchanged in each peptide (*Hoofnagle et al., 2003*). Deuterium exchange for all peptides is represented using relative fractional uptake (RFU) plots. Each value reported is an average of three independent deuterium exchange experiments and not corrected for

back-exchange (*Lim et al., 2017*). Difference plots were made by subtracting absolute centroid mass values between the two states under consideration. A difference of ±0.5 Da was considered a significance threshold for deuterium exchange (*Houde et al., 2011*). Deuteros 2.0 (*Lau et al., 2021*) was used to generate coverage maps and Woods plots with peptide level significance testing.

## Synthesis of fluorescein labeled 11-DOC

Unless noted, materials and solvents were purchased from MilliporeSigma (Burlington, MA) and used without further purification. FAM-DBCO, 6-isomer was purchased from Lumiprobe Corporation (Hunt Valley, MD) and used without further purification. 5-bromovaleryl chloride was purchased from TCI America (Montgomeryville, PA) and used without further purification. Thin layer chromatography (TLC) was performed on Sorbent Technologies XHL 254 silica gel plates. Premium Rf 60 silica gel was used for column chromatography. $^1$H and $^{13}$C NMR spectra were obtained on a Bruker Avance Neo 400 MHz NMR spectromer or Bruker 500 MHz Avance III HD NMR Spectrometer with deuterated solvent as noted. Electrospray ionization (ESI) mass spectrometry was performed using a Thermo Q Exactive mass spectrometer with a Vanquish liquid chromatography system.

## Synthesis

*2-((10 R,13S,17S)–10,13-dimethyl-3-oxo-2,3,6,7,8,9,10,11,12,13,14,15,16,17-tetradecahydro-1H-cyclopenta[a]phenanthren-17-yl)–2-oxoethyl 5-bromopentanoate (2) (Scheme 1).* A solution of 21-hydroxyprogesterone (0.186 g, 0.563 mmol, 1.00 eq), triethylamine (0.157 mL, 1.13 mmol, 2.00 eq), and 4 mL dichloromethane (DCM) in a round bottom flask was cooled to 0 °C. To the solution was added 5-bromovaleryl chloride (0.150 mL, 1.20 mmol, 2.12 eq) dropwise over 5 min. The solution was allowed to warm to room temperature while continuing to stir for 3 hours. The reaction was diluted with more DCM then washed with water and 1 M $K_2CO_3$, dried with $MgSO_4$, filtered, and evaporated. Column chromatography was performed (1:1 Ethyl Acetate:Hexanes, Rf = 0.5) to purify the compound, yielding **2** (0.184 g, 0.373 mmol, 66 %) as a white solid. $^1$H NMR (400 MHz, CDCl$_3$) δ 5.73 (1 H, s), 4.74 (1 H, d, J=17), 4.51 (1 H, d, J=17), 3.42 (2 H, t, J=6.5), 2.52–0.92 (complex, 26 H), 1.17 (3 H, s), 0.69 (3 H, s); $^{13}$C NMR (100 MHz, CDCl$_3$) δ 203.61, 199.73, 172.58, 171.10, 124.06, 69.22, 59.20, 56.29, 53.67, 50.92, 44.79, 38.69, 38.44, 35.80, 35.65, 34.02, 33.24, 32.86, 31.98, 31.90, 24.58, 23.50, 22.96, 21.10, 17.46, 13.30 .

**Scheme 1.** Synthesis of 11-DOC-FAM from 11-deoxycorticosterone.

*2-((10 R,13S,17S)–10,13-dimethyl-3-oxo-2,3,6,7,8,9,10,11,12,13,14,15,16,17-tetradecahydro-1H-cyclopenta[a]phenanthren-17-yl)–2-oxoethyl 5-azidopentanoate (3) (Scheme 1).* To a solution of **2** (0.224 g, 0.454 mmol, 1.00 eq) and 5 mL anhydrous *N,N*-dimethylformamide (DMF) in a round bottom flask, was added sodium azide (0.305 g, 4.69 mmol, 10.3 eq). The solution was was heated to 65 °C

for 16 hours. Upon completion, excess sodium azide was filtered off, and the rest was evaporated. The resulting residue was dissolved in Ethyl Acetate and washed three time with water. The organic phase was dried with $MgSO_4$, filtered, and evaporated. Column chromatography was performed (4:3 Ethyl Acetate:Hexanes, Rf = 0.57) to purify the compound, yielding **3** (0.162 g, 0.356 mmol, 78 %) as a white solid. ¹H NMR (500 MHz, $CD_3CN$) δ 5.66 (1 H, s), 4.76 (1 H, d, J=17), 4.57 (1 H, d, J=17), 3.35 (2 H, t, J=6.5), 2.63–0.98 (complex, 26 H), 1.21 (3 H, s), 0.68 (3 H, s); ¹³C NMR (125 MHz, $CD_3CN$) δ 204.85, 199.46, 173.34, 172.11, 124.09, 70.09, 59.55, 56.82, 54.48, 51.69, 45.23, 39.41, 38.88, 36.47, 36.24, 34.55, 33.63, 33.22, 32.78, 28.71, 25.04, 23.34, 22.80, 21.73, 17.61, 13.44 .

*2-((10 R,13S,17S)–10,13-dimethyl-3-oxo-2,3,6,7,8,9,10,11,12,13,14,15,16,17-tetradecahydro-1H-cyclopenta[a]phenanthren-17-yl)–2-oxoethyl 5-(8-(6-(3',6'-dihydroxy-3-oxo-3H-spiro[isobenzofuran-1,9'-xanthene]–6-carboxamido)hexanoyl)–8,9-dihydro-1H-dibenzo[b,f](1,2,3)triazolo[4,5-d]azocin-1-yl) pentanoate* (**4**) . A solution of **3** (0.027 g, 59.3 μmol, 1.22 eq), **FAM-DBCO** (0.033 g, 48.8 μmol, 1.00 eq), 6 mL DCM, and 3 mL methanol were combined in a round bottom flask and stirred at room temperature for 18 hr. The solution was evaporated and the compound was purified by reverse phase preparative HPLC (Agilent Technologies) using a ramp of 0% to 100% B over 10 min (retention time: 6.18 min) to yield **4** (0.009 g, 7.95 μmol, 13 %) as a yellow solid. ESI-MS *m/z* [M+H]⁺ observed: 1132.5, calculated: 1132.5.

## Ensemblator analysis

For structural comparison of WT and mutant ensembles, 100 conformations were obtained from each trajectory, based on combined clustering results. Structures selected represent the lowest RMSD members of the most populated WT or mutant clusters. Subgroups were identified as WT versus mutant groups. Briefly, Ensemblator performs local conformation comparisons by calculating a local overlaid dipeptide residual (LODR) score to measure residue-level backbone similarity (*Brereton and Karplus, 2018*; *Clark et al., 2015*). Global comparisons are performed following a least-squares overlay of all structures using common atoms. From both local and global comparisons, a discrimination index is calculated to access the significance of differences for each atom in both groups. Inter-subgroup variations are calculated as well as intra-subgroup comparisons. The DI is calculated for each atom as the mean of the pairwise distances between the groups minus the mean of the pairwise distances within the group, divided by the higher of the two values. Values range between 0 and 1, going from indistinguishable to structurally distinct ensembles.

## Acknowledgements

The authors wish to thank Dr. Suzanne G Mays for critical reading and insightful comments on this manuscript. We also thank Dr. Andrew Karplus for in-depth discussion and guidance with the Ensemblator tool.

## Additional information

### Funding

| Funder | Grant reference number | Author |
| --- | --- | --- |
| Burroughs Wellcome Fund | | C Denise Okafor |

The funders had no role in study design, data collection and interpretation, or the decision to submit the work for publication.

### Author contributions

Sabab Hasan Khan, Conceptualization, Data curation, Formal analysis, Investigation, Methodology, Writing – original draft, Writing – review and editing; Sean M Braet, Data curation, Formal analysis, Investigation, Methodology, Writing – review and editing; Stephen John Koehler, Elizabeth Elacqua, Resources, Investigation, Methodology, Writing – review and editing; Ganesh Srinivasan Anand, Data

curation, Formal analysis, Methodology, Writing – review and editing; C Denise Okafor, Conceptualization, Data curation, Formal analysis, Supervision, Funding acquisition, Investigation, Visualization, Methodology, Writing – original draft, Project administration, Writing – review and editing

### Author ORCIDs
Sabab Hasan Khan http://orcid.org/0000-0001-5029-2337
Stephen John Koehler http://orcid.org/0000-0002-7888-7663
Ganesh Srinivasan Anand http://orcid.org/0000-0001-8995-3067
C Denise Okafor http://orcid.org/0000-0001-7374-1561

### Decision letter and Author response
Decision letter https://doi.org/10.7554/eLife.80140.sa1
Author response https://doi.org/10.7554/eLife.80140.sa2

---

## Additional files

### Supplementary files
• Supplementary file 1. Oligonucleotide primers used for site-directed mutagenesis.
• Supplementary file 2. Fractional populations for clusters.
• MDAR checklist

### Data availability
HDX data generated during this study has been uploaded to the PRIDE exchange on ProteomeXchange with identifier PXD036076.

The following dataset was generated:

| Author(s) | Year | Dataset title | Dataset URL | Database and Identifier |
|---|---|---|---|---|
| Khan SH, Braet SM, Koehler S, Elacqua E, Anand G, Okafor CD | 2022 | Ligand-induced shifts in conformational ensembles that predict transcriptional activation | https://ebi.ac.uk/pride/PXD036076 | PRIDE, PXD036076 |

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
