## [Editor Report]

This paper reports a fundamental set of results describing the activation of nuclear receptors. The evidence to support the relationship between function and ligand-induced shift in the conformational ensembles is based on a compelling combination of experimental and computational approaches. The manuscript has implications for fully understanding how perturbation of the conformational ensembles of proteins, in general, orchestrates function. The findings will be of interest to a broad audience in biochemistry and structural, molecular, and evolutionary biology.

---

## [Decision Letter]

**Decision letter after peer review:**

Thank you for submitting your article "Ligand-induced shifts in conformational ensembles that predict transcriptional activation" for consideration by *eLife*. Your article has been evaluated by 3 peer reviewers, and this evaluation has been overseen by Donald Hamelberg as Reviewing Editor and José Faraldo-Gómez as Senior Editor. The following individual involved in the review of your submission has agreed to reveal their identity: Argyris Politis (Reviewer #3).

Essential revisions:

1) M75I mutant. The authors' functional studies showing the M75I mutant does not bind ligand are presented after the authors present ligand-bound simulation analyses of M75I and the other mutants vs. WT AncSR2. The significance of the ligand-bound M75I computational simulations is unclear if the ligand does not bind experimentally.

2) Potential for correlation bias. It seems much of the computation-function correlation analyses are performed via visual inspection, which can lead to bias. Computation-activity correlation coefficients, e.g., by plotting the fraction of the simulation in cluster X (or Y or Z) vs. maximum activity of the receptor (without ligand or at 1 µM ligand), would help reduce bias and better support the conclusion that computation methods can predict ligand activity.

3) Computational analysis details and new extended analyses. There are a few points related to how the analyses were performed. It is not entirely clear which simulation figure panels correspond to the analysis of conventional vs. accelerated MD trajectories. The cluster analysis of the simulations in Figure 2 compares two states (e.g., WT vs. M75A; M75 apo vs. EST-bound; etc.). However, a more comprehensive cluster analysis-where WT and the four M75 mutants are all clustered together to discover conserved/unique clusters that are populated in all/some/unique conditions-may better inform computational-function correlations. In the cluster analyses in Figure 2D and E, there are conserved clusters but the fractional population sizes can differ (e.g., M75A in cluster 1 is 1/3 the size as WT); the significance of these differences is not clear. Other features of the plots are not well defined, including the fraction of the total frames represented by the plots that show only several clusters; and the underlying structural similarities and differences among the various clusters when comparing different mutants or liganded states. In Figure 6A and B, it is not clear how frequently interactions are populated in the WT vs. mutant simulations (some information is provided in the manuscript text, but a fractional occupancy plot would help).

4) Constitutive activity. Differences in luciferase transcriptional reporter data are described indicating two mutants (M75L and M75I) display constitutive activity, the structural basis of which is further suggested by MD simulation data. However, an alternative explanation could be the mutants display different expression or protein levels in cells.

5) Limitations. One limitation of the study is the assumption that the ligand-free LBD structural state is similar to the LBD conformation where the PROG-bound AncSR2 LBD was stripped of the ligand (perhaps with some additional simulation to relax the system). However, this may not be representative of the true ligand-free LBD conformational ensemble. Describing the ligand-stripped conformational analyses as inactive may therefore be better described as ligand-free. Related is the assumption that the other ligand-bound states (EST, etc.) are not significantly different from the PROG state or can be accessed after further accelerated MD, which may be more plausible.

6) With a relatively small set of information as presented in this manuscript, would it be better to state that the computational studies predict activity or describe the activity (because in some cases computation does not predict or describe activity; ligand-bound M75I mutant). To predict activity, it could be argued that a larger dataset would be needed perhaps with data training + machine learning.

7) Western blot analysis would inform whether mutant receptor activity differences are caused by differences in protein levels.

8) Manuscript organization. The manuscript story is framed to ask the question if ligand activity be predicted from an experiment with an early emphasis on simulation, then relating the observed/discovered simulation findings to the experiment, then back to simulation. One wonders if the data were presented in a different way and if additional correlation analyses were performed on the existing data if the resulting outcomes would seem less biased and more informed (along with additional analyses and clarification of current analyses described below).

9) The flow of the manuscript may be improved if the manuscript were reorganized to describe the luciferase assay data and ligand displacement data first, then the computational clustering and correlation to experiment-rather presenting the functional studies in between two different computational analyses. A reorganized flow may address a few unanswered questions or speculations made in the current manuscript:

10) If the manuscript was reorganized as described above, one might not choose to include the M75I simulation data or use it to determine if ligand affinity can be predicted from simulation (currently the manuscript only attempts to predict ligand efficacy).

11) On page 5, the authors speculate that H3-H5 distance might impact transcriptional activity but provide no underlying basis for this hypothesis. If this idea has been previously suggested and supported by data, citations should be added. However, presenting the functional data first, then describing these H3-H5 simulation distance findings, would provide an opportunity to state (whether) there is a correlation between this distance and transcriptional output.

12) Another speculative statement where the basis for the comment is not well supported includes: end of the Figure 2 legend, in that the new M75A/M75I ligand-bound conformational states would activate the receptor or not (how can this be inferred from simulation data alone?).

13) Previous work in other nuclear receptors has shown that decreased HDX in the coregulator binding region is associated with receptor activation; however, the authors see the opposite here. M75L has increased HDX compared to WT, apparently imparting constitutive activity. This should be at least pointed out and discussed in the paper.

14) Given the large reduction in Tm for some of the mutants, I am not sure that the Tm reductions are "reflective of local, structural effects". What evidence or rationale do you have that argues/shows that these Tm reductions are not reflective of global changes to the protein?

15) "M75F shows the largest H3-H5 distance (7.5-8.1 Å) while M75L and WT AncSR2 show intermediate distances (7.4-8.1 Å). Thus, we determined that while M75 mutations preserve the contact with hormones, they vastly modulate the H3-H5 interhelical distance which might impact transcriptional activity." The stated numbers don't indicate to me that M75F, M75L, and WT induce different H3-H5 distances.

16) "Strikingly, we note that the largest global changes in M75L coincide with the regions predicted by HDX-MS to be destabilized by the M75L mutation (Figure 5B)." Contrary to this statement, I can't see a significant correlation between Figure 5B (M75L), and Figure 6E (M75L). This calls into question the correlation between the presented HDX-MS and simulation data.

17) How was simulation convergence of the relevant structural states tested? Please include some measure of convergence.

18) Could you display the data differently for Figure 2 panels E and F? It is hard to understand the similarity or difference between apo or wt. Are the wt/apo stacked or overlayed on the mutant/ligand bars? Possible ways to improve clarity are to display the data separately, use unfilled bars, or explicitly show and state that the data are stacked or overlayed.

19) Why do WT, M75F, and M75A produce less luciferase than empty vector (Figure 3F)? The similar supplementary figure suggests that the Y-axis numbers are wrong in Figure 3F.

20) Inclusion of additional data where data not shown is specified, could help consolidate the text. It would have been interesting to include M75A and M75F mutants in the binding assay and HDX to further strengthen the conclusions.

21) Figure S1; the SEC profiles of the WT and the mutants could be included in the same figure (with SDS-PAGE) instead of data not shown. It would help to further strengthen the text.

22) Figure 1D; the decrease in thermal stability of M75I mutant may lead to increased aggregation. Again, it would be interesting to correlate with the SEC profile for a better understanding of the results.

23) Figure 2; the text in the results is difficult to correlate with figures 2C and 2D. Specifically, for M75A and M75I, the H3-H5 distance is specified to be 7-7.2 Å in the text, which is to be referred to in figure 2D, but in figure 2D it is different.

24) Figure 4; the legends C and D are substituted for D and E. Also, legend 'E' is not in bold.

25) No statement is made regarding data availability. The HDX and other data generated during this work are recommended to be made accessible to the public.

26) Amino acid residue callouts use a mixture of three- and one-letter residue callouts, e.g., Met75 and M75; one should be chosen, and if one-letter code is chosen, then helical callouts such as H7 should be renamed h7 or helix7 for clarity.

27) There may be an overuse of words such as "unexpectedly", "unexpected", "vastly", "strikingly", "dramatic", etc.

28) Figure 2B: what atom(s) were used to measure the distance between the ligand and res75-or was a centroid used?

29) In several places, references are needed to support statements such as "our previous work", etc.

30) Maybe instead of using the phrase "conformational shifts" the phrase "a shift of the conformational ensemble" would be more appropriate?

31) Statements such as "none of the hormones activated mutant X" should probably be clarified with an additional phrase like "with ligand treatment up to 1 µM".

32) Page 9, the first line of the paragraph: it may not be unexpected that receptor mutations might inhibit or change activity.

---

## [Author Response]

Essential revisions:1) M75I mutant. The authors' functional studies showing the M75I mutant does not bind ligand are presented after the authors present ligand-bound simulation analyses of M75I and the other mutants vs. WT AncSR2. The significance of the ligand-bound M75I computational simulations is unclear if the ligand does not bind experimentally.

We have reorganized the manuscript based on suggestions from Reviewers. With this reorganization, we agree that this makes the M75I simulations less relevant. We have removed them from the accelerated MD/clustering section but retain the classical MD simulations as these provide a structural rationale for why the mutant is unable to bind ligands.

2) Potential for correlation bias. It seems much of the computation-function correlation analyses are performed via visual inspection, which can lead to bias. Computation-activity correlation coefficients, e.g., by plotting the fraction of the simulation in cluster X (or Y or Z) vs. maximum activity of the receptor (without ligand or at 1 µM ligand), would help reduce bias and better support the conclusion that computation methods can predict ligand activity.

This is an excellent suggestion, and we have now included new correlation plots (Figure 6F-H) to support the observations which were previously purely visual. We have also expanded our original accelerated MD analysis from 5 liganded complexes to 20 by including ligands for each of the receptor variants, adding more confidence and data points to our correlation analysis.

3) Computational analysis details and new extended analyses. There are a few points related to how the analyses were performed. It is not entirely clear which simulation figure panels correspond to the analysis of conventional vs. accelerated MD trajectories.

We have now edited both text and figure captions to explicitly identify data that originate from accelerated MD vs classical MD trajectories.

The cluster analysis of the simulations in Figure 2 compares two states (e.g., WT vs. M75A; M75 apo vs. EST-bound; etc.). However, a more comprehensive cluster analysis-where WT and the four M75 mutants are all clustered together to discover conserved/unique clusters that are populated in all/some/unique conditions-may better inform computational-function correlations.

This is a very interesting point! While it is outside the scope of this study to discover/structurally characterize conformational states that are common across variants, we agree that it would be interesting to predict whether any conformational states are populated across multiple complexes. We proceeded to cluster all complexes together by ligand, excluding M75I liganded complexes. However because we are dealing with a substantially larger number of frames (125000 vs 50000), we are unable to use the MMTSB algorithm and had to employ the k-means clustering algorithm in CPPTRAJ of AmberTools. We also constrained the number of clusters to 10 for each set of complexes to maintain reasonable sizes. These results are interesting, showing that while no conformational states are shared (in significant populations) across all variants, there are varying amounts of overlap observed in the different ligands. Because this has yielded new ideas that will require more time to explore, we would like to study these effects in closer detail and as such, have chosen not to include this data in this manuscript.

In the cluster analyses in Figure 2D and E, there are conserved clusters but the fractional population sizes can differ (e.g., M75A in cluster 1 is 1/3 the size as WT); the significance of these differences is not clear.

Because our intention is to analyze conformational overlap, it is not as important to standardize sizes/numbers of clusters across complexes. To achieve this kind of uniformity, we would have to modify the clustering radius for each complex. We believe this would ultimately yield the same (or very similar) overall trends/overlap fractions. For example, the clustering presented in Figure 6E was performed using clustering radius 2.4, while in our previous work (Okafor et al., Structure 2020), we clustered the same MD simulations using a radius of 2.5. While the numbers/sizes of clusters are different, the overlap fractions are consistent between both analyses.

Additionally, we are focused on achieving an unbiased analysis which could be affected if we were to modify the radius for each complex with the intent of standardizing the number of clusters. We have added the sentence below in the Methods to help explain this rationale:

“A 2.4 Å cutoff was used for all systems, which results in distinct numbers/sizes of clusters for each complexes but allows for an unbiased analysis.”

Other features of the plots are not well defined, including the fraction of the total frames represented by the plots that show only several clusters; and the underlying structural similarities and differences among the various clusters when comparing different mutants or liganded states.

We have made several additions to enhance our clustering analysis.

I. We have included a Supplemental table (Table S2) that reports the fraction of the total population of frames represented by each cluster shown in Figure 6B-E.

II. While it is beyond the goals of this study to discuss the conformational features (differences, similarities) in every cluster produced here, we agree that some structural description would be useful. To this end, we have calculated RMSFs for each cluster and added this data in Figure S12. This allows us to view the most variable regions in each complex, i.e. the areas most likely to differ between clustered conformational states. We have included the sentences below in Discussion to describe this addition.

To quickly identify the most distinguishing features of each cluster from Figure 6B-E, we calculated root mean square fluctuations (RMSF) for all cluster populations (Figure S12). While these reveal that flexible regions of AncSR2 variants largely drive conformational differences between clusters, they also identify residues 100-125, i.e. H6 and H7 as a region that distinguishes clusters for all complexes, most prominently in WT AncSR2 complexes (Figure S12).

In Figure 6A and B, it is not clear how frequently interactions are populated in the WT vs. mutant simulations (some information is provided in the manuscript text, but a fractional occupancy plot would help).

These are now Figures 5E-F. For Figure 5E, we have added a supplemental figure (Figure S9) illustrating that both residues (Q41, W71) are closer to the ligand in the M75A simulation compared to WT. For Figure 5F, we have added a plot showing the distance and angle between the aromatic rings of F75 and W71, used to estimate the frequency of pi-stacking (Figure S11). We have also included these figure references in the text.

4) Constitutive activity. Differences in luciferase transcriptional reporter data are described indicating two mutants (M75L and M75I) display constitutive activity, the structural basis of which is further suggested by MD simulation data. However, an alternative explanation could be the mutants display different expression or protein levels in cells.

We thank the Reviewer for bringing up this point. We have now added the suggested western blot experiments (Figure S3B) which show that expression levels of mutants are comparable to WT AncSR2 protein.

5) Limitations. One limitation of the study is the assumption that the ligand-free LBD structural state is similar to the LBD conformation where the PROG-bound AncSR2 LBD was stripped of the ligand (perhaps with some additional simulation to relax the system). However, this may not be representative of the true ligand-free LBD conformational ensemble. Describing the ligand-stripped conformational analyses as inactive may therefore be better described as ligand-free. Related is the assumption that the other ligand-bound states (EST, etc.) are not significantly different from the PROG state or can be accessed after further accelerated MD, which may be more plausible.

This is an astute observation on the part of the Reviewer. Even though crystal structures of several apo nuclear receptors show the same structural state as the ligand-bound form, it is also known from NMR studies that H12, for instance, is conformationally flexible in the unliganded state. And so it is very difficult to know for sure whether the true apo/inactive state is the same as prog-bound LBD minus ligand. We have removed the use of the word ‘inactive’ to describe conformational states and instead describe them as ligand-free/unliganded.

6) With a relatively small set of information as presented in this manuscript, would it be better to state that the computational studies predict activity or describe the activity (because in some cases computation does not predict or describe activity; ligand-bound M75I mutant). To predict activity, it could be argued that a larger dataset would be needed perhaps with data training + machine learning.

We agree with this statement, and we have changed references stating that MD ‘predicts’ the activity of receptors to saying that it describes these data. We have also adjusted the title to replace ‘predict’ with ‘describe’.

7) Western blot analysis would inform whether mutant receptor activity differences are caused by differences in protein levels.

We have performed the proposed western blot experiments (Figure S3B) which shows expression levels of all mutants are comparable to WT AncSR2.

8) Manuscript organization. The manuscript story is framed to ask the question if ligand activity be predicted from an experiment with an early emphasis on simulation, then relating the observed/discovered simulation findings to the experiment, then back to simulation. One wonders if the data were presented in a different way and if additional correlation analyses were performed on the existing data if the resulting outcomes would seem less biased and more informed (along with additional analyses and clarification of current analyses described below).

Please see the next answer for our response

9) The flow of the manuscript may be improved if the manuscript were reorganized to describe the luciferase assay data and ligand displacement data first, then the computational clustering and correlation to experiment-rather presenting the functional studies in between two different computational analyses. A reorganized flow may address a few unanswered questions or speculations made in the current manuscript:

We have taken this suggestion to heart and reorganized the manuscript. We now present all experimental data first, followed by computational studies with clustering geared at demonstrating the ability of simulations to predict/describe transcriptional activity in various mutant-ligand combinations.

10) If the manuscript was reorganized as described above, one might not choose to include the M75I simulation data or use it to determine if ligand affinity can be predicted from simulation (currently the manuscript only attempts to predict ligand efficacy).

With the re-organization of this manuscript based on suggestions from Reviewers, we agree that this makes the M75I simulations less relevant. We have removed them from the accelerated MD/clustering section but retain the classical MD simulations as these provide a structural rationale for why the mutant is unable to bind ligands.

11) On page 5, the authors speculate that H3-H5 distance might impact transcriptional activity but provide no underlying basis for this hypothesis. If this idea has been previously suggested and supported by data, citations should be added. However, presenting the functional data first, then describing these H3-H5 simulation distance findings, would provide an opportunity to state (whether) there is a correlation between this distance and transcriptional output.

While this is the first time (to our knowledge) that a correlation is proposed between H3-H5 distance and activity, there is previous demonstration of the importance of a conserved H3-H5 contact in steroid receptors. However the Reviewer is correct in that more work would be needed to support the hypothesis we present here. We have changed the confident nature of this statement to instead suggest that this is a potential hypothesis that should be considered/tested in the future.

12) Another speculative statement where the basis for the comment is not well supported includes: end of the Figure 2 legend, in that the new M75A/M75I ligand-bound conformational states would activate the receptor or not (how can this be inferred from simulation data alone?).

This statement was based on the described hypothesis that based on the lack of conformational overlap, both receptors would be predicted to be activated by estrogen and progesterone. However as we have reorganized the manuscript and now present the computational data after the experiments, this statement has been removed.

13) Previous work in other nuclear receptors has shown that decreased HDX in the coregulator binding region is associated with receptor activation; however, the authors see the opposite here. M75L has increased HDX compared to WT, apparently imparting constitutive activity. This should be at least pointed out and discussed in the paper.

This is an interesting point, because we do not observe any changes near the AF-2 surface in our M75L-WT HDX experiments. Instead, we observe increases in (H3, H10) and adjacent to the binding pocket (H6, H7), as well as distant locations such as H9. Changes near AF-2 would suggest that perhaps M75L stabilizes conformations of AncSR2 that more easily recruit coregulators, which might explain the constitutive activity. However, changes in the region we observe suggest that the mutation impacts ligand binding. One plausible explanation is that the M75L enables the binding pocket to sample a range of conformations, including some resembling ligand-bound states which may permit constitutive activity. We have edited this portion of the text as follows.

An increase in deuterium uptake was observed in multiple regions of the M75L mutant, including residues in and adjacent to the ligand binding pocket: H3, H10, H6 and H7. This deprotection may indicate general destabilization of these LBD regions. However the localization of these changes to the binding pocket may also suggest that M75L allows the pocket to sample a range of conformations, including some that resemble ligand-bound states which may permit constitutive activity. Deprotection is also unexpectedly observed in distant regions such as H9 and the N-terminal end of H10.

14) Given the large reduction in Tm for some of the mutants, I am not sure that the Tm reductions are "reflective of local, structural effects". What evidence or rationale do you have that argues/shows that these Tm reductions are not reflective of global changes to the protein?

We rationalized this statement based on our Far-UV circular dichroism studies. The far-UV CD spectra of less stable mutants overlapped with WT AncSR2, suggesting similar global conformations in our experimental conditions. In addition, gel filtration profiles of all mutants show that LBDs elute at the same position as WT AncSR2, confirming that mutants maintain similar compactness as AncSR2. However, we cannot rule out the possibility of differences in tertiary structure which may potentially play significant roles in the stabilization of the folded protein.

15) "M75F shows the largest H3-H5 distance (7.5-8.1 Å) while M75L and WT AncSR2 show intermediate distances (7.4-8.1 Å). Thus, we determined that while M75 mutations preserve the contact with hormones, they vastly modulate the H3-H5 interhelical distance which might impact transcriptional activity." The stated numbers don't indicate to me that M75F, M75L, and WT induce different H3-H5 distances.

Thank you for this observation! We have changed this sentence to more reflect the average/largest distances observed instead of the range, which makes the differences more obvious. The first sentence now reads as below:

M75F shows the largest H3-H5 distances (as high as 8.1 Å between Leu42-Trp71) while M75L and WT AncSR2 show intermediate distances on average.

16) "Strikingly, we note that the largest global changes in M75L coincide with the regions predicted by HDX-MS to be destabilized by the M75L mutation (Figure 5B)." Contrary to this statement, I can't see a significant correlation between Figure 5B (M75L), and Figure 6E (M75L). This calls into question the correlation between the presented HDX-MS and simulation data.

This statement was specifically in reference to the large changes in the H9-H10 loop as well as H6/H7 which are both observed in HDX-MS. We have changed this statement to specifically highlight these two regions which are unexpected observations as they are not directly in the ligand binding pocket. This sentence has been modified as follows:

Interestingly, we note that some of the large global changes in M75L (H6, H7, H9) coincide with regions predicted by HDX-MS to be destabilized by the M75L mutation (Figure 4B).

17) How was simulation convergence of the relevant structural states tested? Please include some measure of convergence.

As the Reviewer may appreciate, it is difficult to reach true convergence with the 500-ns trajectories performed here. It is important to clarify that our goal in this work is not to reach convergence. Rather, by performing triplicate simulations (for classical MD) and in using enhanced sampling/accelerated MD, we sought to maximize conformational sampling within the limits of our abilities, anticipating that this would improve our likelihood of extracting physiologically relevant details from these systems.

However to assess structural stability and gain some sense of equilibration, we have calculated RMSDs over all trajectories. These figures are added in Figure S8.

18) Could you display the data differently for Figure 2 panels E and F? It is hard to understand the similarity or difference between apo or wt. Are the wt/apo stacked or overlayed on the mutant/ligand bars? Possible ways to improve clarity are to display the data separately, use unfilled bars, or explicitly show and state that the data are stacked or overlayed.

We thank the Reviewer for suggestions to improve this figure. We have updated the captions (Figure 6) and text to explicitly indicate that bars are stacked, as well as provided fractional populations for each cluster in the SI (Table S2).

19) Why do WT, M75F, and M75A produce less luciferase than empty vector (Figure 3F)? The similar supplementary figure suggests that the Y-axis numbers are wrong in Figure 3F.

All of our constructs for luciferase contain the AncSR2 variant ligand binding domains (LBDs) fused to a Gal4DBD, while our empty vector contains the Gal4DBD alone. Previous work has shown that in the absence of hormone, the Gal4DBD alone can have minor luciferase activity which may be repressed by the presence of the LBD (https://pubmed.ncbi.nlm.nih.gov/19014486/). This transrepression was overcome by very low hormone concentrations (1 nM). This past observation provides a plausible explanation for the lower luciferase activity observed in WT/M75F/M75A, as well as support the presence of constitutive activity in M75L and M75I which both retain activity in the absence of ligand.

Regarding the comparison between Figure 3F (now 2F) and Figure S3A, as these are performed in different cell types, it is not typical to compare the fold activation/changes directly between them. The most appropriate comparison is to check that there is a statistically significant change in fold activation in M75L/M75I compared to all other variants, which is consistent across both.

20) Inclusion of additional data where data not shown is specified, could help consolidate the text. It would have been interesting to include M75A and M75F mutants in the binding assay and HDX to further strengthen the conclusions.

We thank the Reviewer for this suggestion and have now included M75A and M75F mutants in the binding assay to provide comparisons for how these mutations affect ligand binding (Figure 3B).

21) Figure S1; the SEC profiles of the WT and the mutants could be included in the same figure (with SDS-PAGE) instead of data not shown. It would help to further strengthen the text.

We have now included these data as Figures S1B and S1C in the SI.

22) Figure 1D; the decrease in thermal stability of M75I mutant may lead to increased aggregation. Again, it would be interesting to correlate with the SEC profile for a better understanding of the results.

We agree with the Reviewer’s suggestions. The M75I gel-filtration profile shows that it remains in the soluble form and eluted similarly to WT AncSR2 and other mutants. The manuscript has been revised to state this explicitly, and gel filtration chromatograms have been included in Figure S3B.

23) Figure 2; the text in the results is difficult to correlate with figures 2C and 2D. Specifically, for M75A and M75I, the H3-H5 distance is specified to be 7-7.2 Å in the text, which is to be referred to in figure 2D, but in figure 2D it is different.

Thank you for this observation, we have made adjustments to make sure the captions, figures (now Figure 5) and text are consistent.

24) Figure 4; the legends C and D are substituted for D and E. Also, legend 'E' is not in bold.

Thank you for bringing this to our attention! We have carefully gone through and fixed this figure legend.

25) No statement is made regarding data availability. The HDX and other data generated during this work are recommended to be made accessible to the public.

The raw data file collected for HDX-MS is now uploaded on the ProteomeX change database (PXD036076). This data is publicly available, and the link is shared in the revised manuscript.

26) Amino acid residue callouts use a mixture of three- and one-letter residue callouts, e.g., Met75 and M75; one should be chosen, and if one-letter code is chosen, then helical callouts such as H7 should be renamed h7 or helix7 for clarity.

We have changed all references in the text to use the 3 letter codes, except for any references to M75 mutations (i.e. M75A, M75F, M75I, M75L) since these occur too frequently throughout the manuscript. However for clarity, we define the first use of M75 in parenthesis next to Methionine 75.

27) There may be an overuse of words such as "unexpectedly", "unexpected", "vastly", "strikingly", "dramatic", etc.

We have significantly reduced the use of these words.

28) Figure 2B: what atom(s) were used to measure the distance between the ligand and res75-or was a centroid used?

We have updated the methods with the text below to clarify how these distances were calculated.

To identify contacts, the ‘distance’ command of CPPTRAJ was used to measure the distance between all pairs of heavy atoms (i.e. non-H atoms) on both residues. For each frame in the analyses, the shortest pair distance was recorded and these values averaged at the end to obtain the miminum distance between heavy atoms.

29) In several places, references are needed to support statements such as "our previous work", etc.

We have added this reference in multiple places and thank the Reviewer for the observation.

30) Maybe instead of using the phrase "conformational shifts" the phrase "a shift of the conformational ensemble" would be more appropriate?

We thank Reviewer for the suggestion. We have removed most occurrences of this phrase and replaced with the Reviewer suggestion.

31) Statements such as "none of the hormones activated mutant X" should probably be clarified with an additional phrase like "with ligand treatment up to 1 µM".

We have incorporated the suggestion in the revised manuscript.

32) Page 9, the first line of the paragraph: it may not be unexpected that receptor mutations might inhibit or change activity.

With the rearrangement of the manuscript and the extra experimental data added, this text is now eliminated.